# Mutation of key signaling regulators of cerebrovascular development in vein of Galen malformations

To elucidate the pathogenesis of vein of Galen malformations (VOGMs), the most common and most severe of congenital brain arteriovenous malformations, we performed an integrated analysis of 310 VOGM proband-family exomes and 336,326 human cerebrovasculature single-cell transcriptomes. We found the Ras suppressor p120 RasGAP (*RASA1*) harbored a genome-wide significant burden of loss-of-function de novo variants (2042.5-fold, $p = 4.79 \times 10^{-7}$). Rare, damaging transmitted variants were enriched in Ephrin receptor-B4 (*EPHB4*) (17.5-fold, $p = 1.22 \times 10^{-5}$), which cooperates with p120 RasGAP to regulate vascular development. Additional probands had damaging variants in *ACVRL1*, *NOTCH1*, *ITGB1*, and *PTPN11*. *ACVRL1* variants were also identified in a multi-generational VOGM pedigree. Integrative genomic analysis defined developing endothelial cells as a likely spatio-temporal locus of VOGM pathophysiology. Mice expressing a VOGM-specific *EPHB4* kinase-domain missense variant (Phe867Leu) exhibited disrupted developmental angiogenesis and impaired hierarchical development of arterial-capillary-venous networks, but only in the presence of a "second-hit" allele. These results illuminate human arterio-venous development and VOGM pathobiology and have implications for patients and their families.

Cerebrovascular system development is a complex, genetically determined process governed by spatially and temporally coordinated events required to meet the hemodynamic and nutritive demands of embryogenesis[1,2]. The developmental changes in gene expression that regulate vasculogenesis, angiogenesis, and arterio-venous specification[3–5] give rise to distinct, contiguous cerebral vessels differentiated into segments identified as arteries, capillaries, and veins[2]. The heterogeneous cell composition along this hierarchically-organized arterio-venous axis includes endothelial cells, pericytes, smooth muscle cells, and other perivascular cells (e.g., neurons, immune cells, and fibroblasts)[1,6,7]. Although experiments in model systems, including chick embryos, zebrafish, and mice, have detailed numerous coordinated molecular interactions within and among these cells that endow the cerebrovasculature with its specialized structural and functional properties, the genetic regulation of arteriovenous development in humans remains poorly understood[2]. The genomic study of rare, severe congenital cerebrovascular anomalies can help elucidate the genes and pathways essential for human cerebrovasculature development, allowing identification of candidate therapeutic targets potentially relevant for more common forms of disease[8,9].

During normal brain development, primitive choroidal and subependymal arteries that perfuse deep brain structures are connected via a primitive meningeal capillary network to the embryonic precursor of the vein of Galen, termed the median prosencephalic vein of Markowski (MPV). The MPV returns deep cerebral venous blood to dural sinuses that drain into the internal jugular veins[10]. Vein of Galen malformations (VOGMs), the most common and severe arteriovenous malformations (AVMs) of the human neonatal brain[11,12], directly connect primitive choroidal or subependymal cerebral arteries to the MPV without an intervening capillary network. This pathological connection exposes the MPV to dangerously high blood flow and blood

✉ e-mail: kingp@umich.edu; jin810@wustl.edu; Kahle.Kristopher@mgh.harvard.edu

pressures that can lead to high-output cardiac failure, hydrocephalus associated with venous congestion, and intracerebral hemorrhage[13] (Fig. 1a). VOGMs can also be associated with other neurodevelopmental pathology and structural heart defects[14], and can be fatal if untreated. Although VOGM treatment has greatly benefited from recent advances in endovascular therapy[15], the morbidity and mortality related to VOGM remain high, even at specialized referral centers[16,17].

Our limited understanding of the molecular pathophysiology of VOGMs has hindered the development of early diagnostic tests and non-procedural therapeutic strategies. Early hypotheses proposed VOGMs were secondary to thrombosis of the straight sinus[18–21]. However, newer theories suggest VOGMs may instead reflect failed angiogenic intussusception during the development of the choroid plexuses, with ischemia playing a potential role[22]. Although most VOGMs are sporadic lesions, VOGMs have also been associated with several Mendelian disorders, sometimes co-existing with other intracranial and extracranial AVMs and multifocal capillary malformations

and telangiectasias[23–26]. These include several cases of autosomal dominant (AD) capillary malformation-AVM syndrome types 1 and 2 (CM-AVM1/2), caused by *RASA1* variants (OMIM: 608354)[27] or *EPHB4* variants (OMIM: 618196)[25], and single VOGM cases in AD hereditary hemorrhagic telangiectasia type 1 and 2 (HHT1/2) caused by variants in *ENG* (OMIM: 187300)[28] or *ACVRL1* (OMIM: 600376)[29]. These data implicate genetic contributions to VOGM, but the underlying genetic cause of most cases remains unknown. Moreover, the cellular and molecular mechanism of VOGM-associated variants are poorly understood due, in part, to lack of adequate mammalian models.

The rare, sporadic nature of VOGM cases[30–32] has limited the power of traditional human genetic approaches (including linkage and genome-wide association studies) to identify causative genes for VOGM and other congenital cerebrovascular lesions. These limitations motivate the curation of deeply-phenotyped proband-parent ("trio") based cohorts, recruited through multi-institutional, international collaboration. Whole-exome sequencing (WES) and statistical analyses can then be applied to search for rare damaging variants in probands

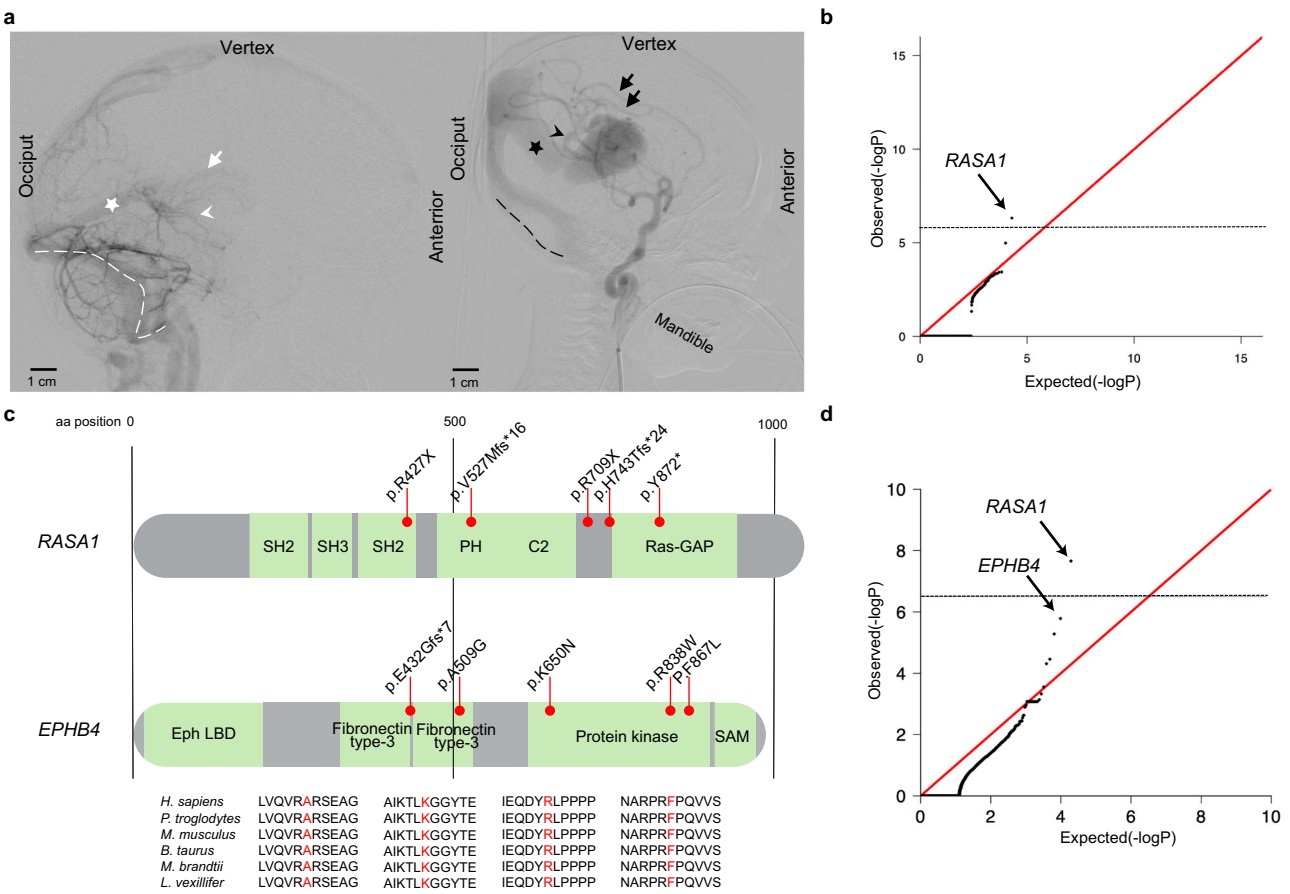

**Fig. 1 | VOGM-associated variants in *RASA1* and *EPHB4*. a** In the normal anatomy shown on a lateral projection subtracted angiogram of a 4-year-old boy (left), the deep veins drain into the straight sinus (white star). The normal deep venous anatomy involves the internal cerebral veins (white arrow) and the basal vein of Rosenthal (white arrowhead). VOGM is shown on a lateral projection subtracted angiogram of a 3 day-old boy (right), where the falcon sinus (black star) is commonly observed as the draining vein and involves prominent supply from splenial vessels deriving from the anterior cerebral artery (black arrows), as well as posterior choroidal arteries (black arrowhead). The normal drainage of the torcular typically involves the bilateral transverse and sigmoid sinuses (white dashed line). In the abnormal venous anatomy, VOGM typically recruit dilated occipital sinuses (black dashed line). **b** Quantile-quantile plot of observed versus expected *p*-values (one-sided Poisson test, not adjusted) for rare damaging (D-mis + LoF) DNVs with MAF ≤ $4 \times 10^{-4}$ in the Exome Aggregation Consortium database for all genes. The

exome-wide significant cutoff was $8.6 \times 10^{-7}$ (0.05 / (3 × 19,347)). D-mis are missense variants predicted to be deleterious per MetaSVM or MPC ≤ 2. LoF, loss-of-function variants. DNV, de novo variant. MAF, minor allele frequency. **c** RASA1 and EPHB4 functional domains (green rectangles) with the location of VOGM variants and phylogenetic conservation of wild-type amino acids (red text) at each mutated position. 5 LoF variants were found in *RASA1*, including 2 DNVs (p. R427X and p. V527Mfs*16), and 3 transmitted variants (p. H743fs*24, p. R709X and p. Y872*). Transmitted damaging variants in *EPHB4* including 2 variants in the Fibronectin type III domain (p. E432Gfs*7 and p. A509G), and 3 D-mis variants (p. K650N, p. R838W, p. F867L) in the kinase domain. **d** Quantile-quantile plots of observed versus expected *p*-values (one-sided Fisher's exact test, not adjusted) for rare damaging (D-mis + LoF) variants with MAF ≤ $5 \times 10^{-5}$ in the Genome Aggregation database (gnomAD) from case-control burden test. The genome-wide significant cutoff was $2.6 \times 10^{-6}$ (0.05 of 19,347).

more often than expected by chance[26]. This agnostic genomic approach has aided gene discovery for other brain and cerebrovascular malformations[33–36], congenital heart disease[37], congenital hydrocephalus[38,39], as well as heterogeneous neurodevelopmental disorders like autism, craniosynostosis, and epilepsy[40–45]. In addition, recent single-cell RNA sequencing (scRNA-seq) studies have begun to systematically catalog cell types of the developing and mature human cerebrovasculature, expanding knowledge about perivascular cell diversity and defining endothelial cell molecular signatures correlated with arteriovenous segmentation[46,47]. Integration of WES findings with other large-scale -omics datasets can help elucidate the cellular and molecular mechanisms of disease genes by defining their spatio-temporal expression patterns and associated transcriptional and protein-protein interaction networks[36,39,48].

Here, we aimed to integrate trio-based WES data of the largest VOGM cohort collected to date (310 proband-family exomes) with 336,326 human cerebrovasculature single-cell transcriptomes in an integrated, systems-level investigation of VOGM pathogenesis. We hypothesized that: (1) multiple novel VOGM candidate genes harboring pathogenic de novo and transmitted variants will be discovered using trio-based WES; (2) VOGM genes will spatiotemporally converge in co-expression modules, cell types, and biological pathways pertinent to the regulation of endothelial biology in ways distinct from those of other congenital cerebrovascular diseases; (3) systematic comparison of phenotypic data from individual VOGM cases will assist gene discovery by clustering cases with similar endophenotypes, thereby defining clinically-relevant disease subclasses; and (4) functional studies of candidate variants in model systems can increase confidence in mutant gene pathogenicity and provide insight into variant mechanisms.

## Results

### WES of the largest trio-based VOGM cohort to date

We ascertained a total of 114 probands with radiographically confirmed VOGM (see Methods) treated by endovascular therapy, including 90 proband-parent trios (each with a single affected offspring), 13 duo cases, and 11 singleton cases (Supplementary Data 1). These included 55 previously described VOGM probands[26]. Among 114 VOGM probands, 34.2% were female; 78.9% were self-reported Europeans; 60% of probands were diagnosed prenatally or within 1 month after birth; only 3.8% were diagnosed after age 2. Salient features at diagnosis included developmental delay (54%), macrocephaly (48%), hydrocephalus (48%), prominent face and/or scalp vasculature (45%), cutaneous vascular lesions (22%), and congestive heart failure (40%). Also present were structural heart defects (7%), including partial anomalous pulmonary venous return, patent ductus arteriosus, and pulmonary valve stenosis. Supplementary Table 1 summarizes the demographics and clinical features of our VOGM cohort. Supplementary Fig. 1 depicts representative angiographic images of VOGM probands.

DNA was isolated and WES was performed (see Methods)[26]. In parallel, WES of 1,798 control trios comprising parents and unaffected siblings of autism probands was analyzed[49,50] by our published informatics pipeline[51]. 92.7% or more of targeted bases had ≥8 independent reads in both cases and controls, and 87.8% or more had ≥15 independent reads (see Supplementary Table 2 for exome sequence metrics). Variant calling was performed utilizing a combination of the Genome Analysis Toolkit (GATK) HaplotypeCaller[52,53] and Freebayes[54]. Population allele frequencies were annotated by the Genome Aggregation Database (gnomAD v.2.1.1) and the BRAVO databases[55]. De novo variant (DNV) identification was performed by TrioDeNovo[56]. MetaSVM and MPC algorithms were used to infer the effect of missense variants[57,58]. Missense variants were considered damaging (D-mis) when predicted as deleterious by MetaSVM or with MPC-score

≥ 2. Inferred loss-of-function (LoF) variants, including stop-gains, stop-losses, frameshift insertions, deletions, and canonical splice-site variants were considered damaging. Variants in genes of interest were validated by PCR amplification and Sanger sequencing (Supplementary Fig. 2 and 4).

### Variants in CM-AVM genes *RASA1* and *EPHB4*

We first investigated the contribution of DNVs to VOGM pathogenesis (Supplementary Data 2). The average DNV rate of 1.19 per subject (Table 1 and Supplementary Table 3) resembled previous results obtained from a similar sequencing platform[39] and followed a Poisson distribution (Supplementary Fig. 5). The burden of DNVs in the control cohort was comparable (Supplementary Table 3). There was no enrichment of synonymous or missense DNVs inferred to be tolerated in VOGM cases. In contrast, D-mis (1.65-fold, one-sided Poisson $p = 0.02$) and protein-damaging DNVs (LoF + D-mis; 1.50-fold, one-sided Poisson $p = 0.02$) were enriched among all genes in cases but not in controls (Table 1). *RASA1* (probability of loss-of-function intolerance [pLI] = 1.00 in gnomAD v.2.1.1) was the single gene that harbors a genome-wide significant burden of damaging DNVs (2042.5-fold, one-sided Poisson $p = 4.79 \times 10^{-7}$) with two novel LoF DNVs (Fig. 1b and Supplementary Table 4). The only other gene with more than one protein-altering DNV (p.Gly202Ser and p.Gln321*) was endothelin-3-converting enzyme *ECE3* (*KEL*) (440.7-fold, one-sided Poisson $p = 1.03 \times 10^{-5}$). From the observed fraction of patients with damaging DNVs we infer that damaging DNVs account for >12% of VOGM cases (Table 1 and see Methods).

*RASA1*, encoding the negative Ras regulator p120 RasGAP[59], contained two novel LoF DNVs, p.Arg427* and p.Val527Mfs*16, in unrelated VOGM probands KVOGM71-1 and KVOGM122-1, respectively (Table 2 and Supplementary Fig. 2). Variants in *RASA1* have been implicated in autosomal dominant type 1 capillary malformation-arteriovenous malformation (CM-AVM1) (OMIM# 608354)[60], featuring fast-flow vascular malformations, including systemic and intracranial arteriovenous fistulas and AVMs and, rarely, VOGMs[27,30,61]. Both KVOGM71-1 and KVOGM122-1 had cutaneous capillary malformations characteristic of CM-AVM type 1. See Supplementary Fig. 6 for available cutaneous manifestations in VOGM probands and their family members.

To identify additional haploinsufficient genes associated with VOGM undetected by DNV analysis, we next assessed the total burden in all probands of rare (BRAVO minor allele frequency [MAF] ≤5 x 10⁻⁵) de novo and transmitted D-mis and LoF variants. The probability of the observed number of rare variants in each gene occurring by chance was calculated by contrasting it with the expected burden, adjusting for gene mutability (see Methods)[39]. Analysis of damaging variants in all genes revealed a genome-wide significant enrichment (Bonferroni multiple testing threshold = 2.6 x 10⁻⁶) of variants in *RASA1*, with five total rare LoF variants (enrichment = 28.4-fold, one-sided binomial $p = 1.20 \times 10^{-6}$; Supplementary Table 5). In addition to the two de novo LoF variants identified in *RASA1* (see above), we identified two transmitted stop-gain variants in *RASA1* (p.Arg709* and p.Tyr695*) in unrelated probands. We also found an unphased frameshift variant in *RASA1* (p.His743Thrfs*24) (Table 2 and Fig. 1c). Case-control burden analysis for rare damaging variants in all probands versus gnomAD controls showed *RASA1* as having a significant mutational burden in VOGM probands (odds ratio (OR) = 67.50 (95% confidence interval [CI]: 25.76, infinite [Inf]), one-sided Fisher's $p = 2.20 \times 10^{-8}$) (Fig. 1d and Supplementary Table 6).

RAS proteins cycle between an active guanosine-triphosphate (GTP)-bound form and an inactive, guanosine-diphosphate (GDP)-bound form, as do other guanine nucleotide-binding proteins (including heterotrimeric G proteins). The weak intrinsic GTPase activity of RAS proteins is greatly enhanced by GTPase-activating

**Table 1 | De novo variant enrichment analysis for each functional class in 90 VOGM trio cases and control cohorts**

**VOGM cases, N = 90**

| | Observed | | Expected | | Enrichment | p |
|---|---|---|---|---|---|---|
| | N | Rate | N | Rate | | |
| **All genes (N = 19,347)** | | | | | | |
| Total | 107 | 1.19 | 99.7 | 1.11 | 1.07 | 0.25 |
| Syn | 28 | 0.31 | 28.2 | 0.31 | 0.99 | 0.54 |
| T-Mis | 46 | 0.51 | 49.4 | 0.55 | 0.93 | 0.71 |
| D-Mis | 22 | 0.24 | 13.4 | 0.15 | 1.65 | 0.02 |
| LoF | 11 | 0.12 | 8.7 | 0.10 | 1.27 | 0.26 |
| Protein altering | 79 | 0.88 | 71.5 | 0.79 | 1.11 | 0.20 |
| Protein damaging | 33 | 0.37 | 22 | 0.24 | 1.5 | 0.02 |
| **Loss-of-function intolerant genes (gnomADv2.1.1 pLI ≥ 0.9; N = 3,049)** | | | | | | |
| Total | 22 | 0.24 | 23.9 | 0.27 | 0.92 | 0.68 |
| Syn | 4 | 0.04 | 6.7 | 0.07 | 0.59 | 0.90 |
| T-Mis | 6 | 0.07 | 10.9 | 0.12 | 0.55 | 0.96 |
| D-Mis | 9 | 0.10 | 4.2 | 0.05 | 2.12 | 0.03 |
| LoF | 3 | 0.03 | 2.1 | 0.02 | 1.4 | 0.36 |
| Protein altering | 18 | 0.20 | 17.2 | 0.19 | 1.05 | 0.45 |
| Protein damaging | 12 | 0.13 | 6.4 | 0.07 | 1.88 | 0.03 |

**Controls, N = 1798**

| | Observed | | Expected | | Enrichment | p |
|---|---|---|---|---|---|---|
| | N | Rate | N | Rate | | |
| **All genes (N = 19,347)** | | | | | | |
| Total | 1839 | 1.02 | 1977.1 | 1.10 | 0.93 | 1.00 |
| Syn | 492 | 0.27 | 559.8 | 0.31 | 0.88 | 1.00 |
| T-Mis | 949 | 0.53 | 979.3 | 0.54 | 0.97 | 0.84 |
| D-Mis | 248 | 0.14 | 266.7 | 0.15 | 0.93 | 0.88 |
| LoF | 150 | 0.08 | 171.3 | 0.10 | 0.88 | 0.95 |
| Protein altering | 1347 | 0.75 | 1417.3 | 0.79 | 0.95 | 0.97 |
| Protein damaging | 398 | 0.22 | 438 | 0.24 | 0.91 | 0.98 |
| **Loss-of-function intolerant genes (gnomADv2.1.1 pLI ≥ 0.9; N = 3,049)** | | | | | | |
| Total | 456 | 0.25 | 473.5 | 0.26 | 0.96 | 0.83 |
| Syn | 115 | 0.06 | 133.4 | 0.07 | 0.86 | 0.96 |
| Mis | 233 | 0.13 | 213.5 | 0.12 | 1.08 | 0.12 |
| D-Mis | 75 | 0.04 | 84.5 | 0.05 | 0.88 | 0.87 |
| LoF | 33 | 0.02 | 42.2 | 0.02 | 0.78 | 0.94 |
| Protein altering | 341 | 0.19 | 340.2 | 0.19 | 1 | 0.53 |
| Protein damaging | 108 | 0.06 | 126.7 | 0.07 | 0.85 | 0.96 |

N Number of de novo variants (DNVs), Rate: number of DNVs per subject; Enrichment: ratio of observed to expected numbers of DNVs; Syn synonymous variants, D-mis damaging missense variants as predicted by MetaSVM or MPC ≥2, T-mis tolerated missense variants as predicted by MetaSVM or MPC <2, LoF loss-of-function variants comprised of premature termination, frameshift, or splice-site variants; Significance threshold determined by upper cumulative quantile (one-sided) of Poisson distribution. Not adjusted.

**Table 2 | Characteristics of Patients With Variants in RASA1 and EPHB4**

| Proband ID | Ethnicity | Sex | Type | Proband Phenotype | Carrier Parent Phenotype | Class | Gene | Position (GRCH37) | AA Change |
|---|---|---|---|---|---|---|---|---|---|
| KVOGM_71-1 | African | F | DNV | Cutaneous vascular lesion | NA | stopgain | RASA1 | 5:86648999:C:T | p.R427X |
| KVOGM122-1 | European | M | DNV | Cutaneous vascular lesion | NA | frameshift | RASA1 | 5:86659286:TTCTG:T | p.V527Mfs*16 |
| KVOGM95-1 | European | F | Transmitted from mother | Deaf left ear | Cutaneous vascular lesions (mother) | frameshift | RASA1 | 5:86672737:T:TC | p.H743Tfs*24 |
| KVOGM42-1* | Mexican | F | Transmitted from mother | Cutaneous vascular lesion, cerebral palsy, neurodevelopmental delay, seizures | Cutaneous vascular lesions (mother) | stopgain | RASA1 | 5:86672323:C:T | p.R709X |
| KVOGM48-1 | European | F | Transmitted from mother | Cutaneous vascular lesion, neurodevelopmental delay | Cutaneous vascular lesions (mother) | stopgain | RASA1 | 5:86676336:T:TA | p.Y695* |
| KVOGM18-1* | European | M | Transmitted from father | Cerebral palsy, hemiplegia, macular scarring, visual neglect, neurodevelopmental delay, seizures | NA | D-mis | EPHB4 | 7:100403202:A:G | p.F867L |
| KVOGM33-1* | Mexican | F | Transmitted from mother | Neurodevelopmental delay | Cutaneous vascular lesions (mother) | D-mis | EPHB4 | 7:100410537:C:A | p.K650N |
| KVOGM25-1* | European | M | Transmitted from father | Sternum excavatum, neurodevelopmental delay | NA | D-mis | EPHB4 | 7:100414876:G:C | p.A509G |
| VOGM115-1* | European | M | Transmitted from father | Neurodevelopmental delay, seizures | Cutaneous vascular lesions (father) | frameshift | EPHB4 | 7:100417179:CCT:C | p.E432Gfs*7 |
| KVOGM_72-1 | EastAsian | M | Transmitted from father | NA | NA | D-mis | EPHB4 | 7:100403289:G:A | p.R838W |

*These patients have been reported in our previous study[26] (PMID: 30578106). Abbreviations: AA amino acid, F Female, M Male, D-Mis deleteriousness of missense variants, DNV de novo variant.

proteins (GAPs) such as p120 RasGAP[62,63]. Similar to other LoF RASA1 variants[64], variants encoding the prematurely terminated mutants p.Arg427*, p.Val527Metfs*16, p.Arg709*, p.Tyr695*, and p.His743Thrfs*24 are all predicted to cause nonsense-mediated mRNA decay and consequently increase downstream Ras/ERK/MAPK signaling activity[65,66].

We also identified enrichment in rare, damaging variants in EPHB4, encoding Ephrin receptor B4 (EphB4), which physically interacts and cooperates with p120 RasGAP to limit Ras activation[67] (enrichment = 17.5-fold, one-sided binomial $p = 1.22 \times 10^{-5}$; Supplementary Table 5). Case-control gene burden analyses for damaging variants in all probands showed a significant variant burden in EPHB4 versus gnomAD controls (OR = 27.39 (95% CI [10.60, Inf]), one-sided Fisher's $p = 1.65 \times 10^{-6}$; Fig. 1d and Supplementary Table 6). EPHB4 variants[25] have been reported in type 2 AD Capillary malformation-arteriovenous malformation (CM-AVM2) (OMIM# 618196). In contrast to RASA1, all but one of the EPHB4 variants were D-mis and transmitted; three of these are absent from ExAC and gnomAD, and two have a MAF of $<1.48 \times 10^{-5}$ in gnomAD (Table 2).

The only detected LoF variant in EPHB4 (p.Glu432Glyfs*7) immediately precedes the fibronectin type 3 (fn3) domain, truncating the protein kinase domain (Fig. 1c), and is predicted to result in nonsense-mediated mRNA decay[68]. All D-mis variants in EPHB4 alter highly conserved amino acid residues (Fig. 1c). p.Lys650Asn and p.Phe867Leu localize to the Eph-B4 tyrosine kinase domain[69,70]. p.Lys650Asn is a surface-exposed residue in the β3-αC loop of the EPHB4 tyrosine kinase N-lobe and is located proximal to the putative binding site of the autoinhibitory juxtamembrane region. p.Phe867Leu is a highly conserved, hydrophobic core residue in the C-lobe of the EPHB4 tyrosine kinase domain. p.Ala509Gly is a highly conserved, hydrophobic core residue in the second of two EPHB4 extracellular fibronectin III domains (Supplementary Fig. 2). In silico biophysical modeling suggests these VOGM-associated D-mis variants significantly disrupt EPHB4 structure and function[26].

The steady-state abundance of Lys650Asn, Arg838Trp, and Phe867Leu EPHB4 variants expressed in Cos-7 cells approximated that of WT EPHB4 (Fig. 2a and Supplementary Fig. 3). Cycloheximide block of protein translation revealed polypeptide decay rates of EPHB4 VOGM mutants similar to that of WT EPHB4 (Fig. 2b), indicating that none of the EPHB4 kinase domain variants affect EPHB4 protein stability. However, none of the VOGM-associated EPHB4 kinase domain mutants contained detectable phosphotyrosine as detected by anti-pTyr immunoblotting of whole-cell lysates and anti-EPHB4 immuno-precipitates (Fig. 2c). These findings strongly suggest that VOGM-associated EPHB4 D-mis variants block the protein tyrosine kinase activity of EPHB4 or render EPHB4 phospho-sites more susceptible to dephosphorylation.

Patient phenotypes associated with RASA1 and EPHB4 variants are described in Supplementary Table 7. These kindreds notably include eight additional family members without diagnosed VOGM who carry the same variants. Among these, three carriers of RASA1 variants and two carriers of EPHB4 variants had cutaneous vascular lesions (Supplementary Table 7). In contrast, cutaneous vascular lesions and cardiac abnormalities were absent among confirmed non-carrier family members. For example, KVOGM42-1 with the rare transmitted RASA1 p.Arg709* variant had a carrier mother without a VOGM, but with cutaneous vascular lesions and an extensive family history on the maternal side of aneurysms, stroke, and Raynaud's syndrome in several other family members unavailable for sequencing. Similarly, the father of KVOGM115-1 carrying the EPHB4 p.Glu432Glyfs*7 variant lacked VOGM but had a capillary malformation. The proband also had a brother with multiple cutaneous capillary malformations and other vascular lesions on his face and leg unavailable for sequencing. Interestingly, none of the VOGM probands or their family members carried a clinical or genetic

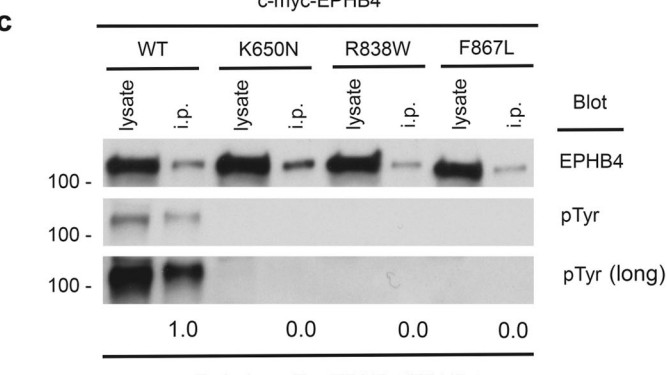

**Fig. 2 | Preserved protein stability with compromised tyrosine kinase activity in EPHB4 kinase domain variants. a** Steady-state abundance of EPHB4 D-mis mutants. Cos-7 cells were transiently transfected with c-myc-tagged WT or EPHB4 D-mis mutants together with an eGFP-encoding vector. Numbers at top indicate percentage of GFP+ cells determined by flow cytometry (see Supplementary Fig. 3). Cells were lysed and EPHB4 abundance determined by Western blot for c-myc (left) or EPHB4 (right). Blots were probed for tubulin or beta actin (ActB) respectively to demonstrate equivalent protein loading. Numbers below indicate normalized abundance of EPHB4 D-mis mutants relative to WT EPHB4. Each experiment was repeated two times with similar results. **b** Stability of EPHB4 D-mis mutants. Cos-7 cells transiently transfected with c-myc-tagged WT or EPHB4 D-mis mutants were treated with cycloheximide (CHX) for the indicated times before lysis and determination of EPHB4 and ActB abundance by Western blot. Numbers indicate normalized EPHB4 abundance relative to CHX-untreated cells for each time point. Shown are the results of a single experiment. **c** Kinase activity of EPHB4 D-mis mutants. EPHB4 was immunoprecipitated from Cos-7 cells transiently transfected with c-myc-tagged WT or EPHB4 D-mis mutants. EPHB4 kinase activity was determined by Western blot of immunoprecipitates with an anti-phosphotyrosine antibody (pTyr). Numbers indicate normalized pTyr content of EPHB4 D-mis mutants relative to WT EPHB4. Shown are the results of one experiment of two repeats.

diagnosis of CM-AVM before inclusion in this study. These findings together show incomplete penetrance and variable expressivity of transmitted variants in the related CM-AVM genes *RASA1* and *EPHB4* in VOGM.

## Variants in *ACVRL1* and other mutation-intolerant Mendelian vascular disease genes

To identify other potential VOGM genes and gain insight into the molecular pathways impacted by their variant, we performed Gene

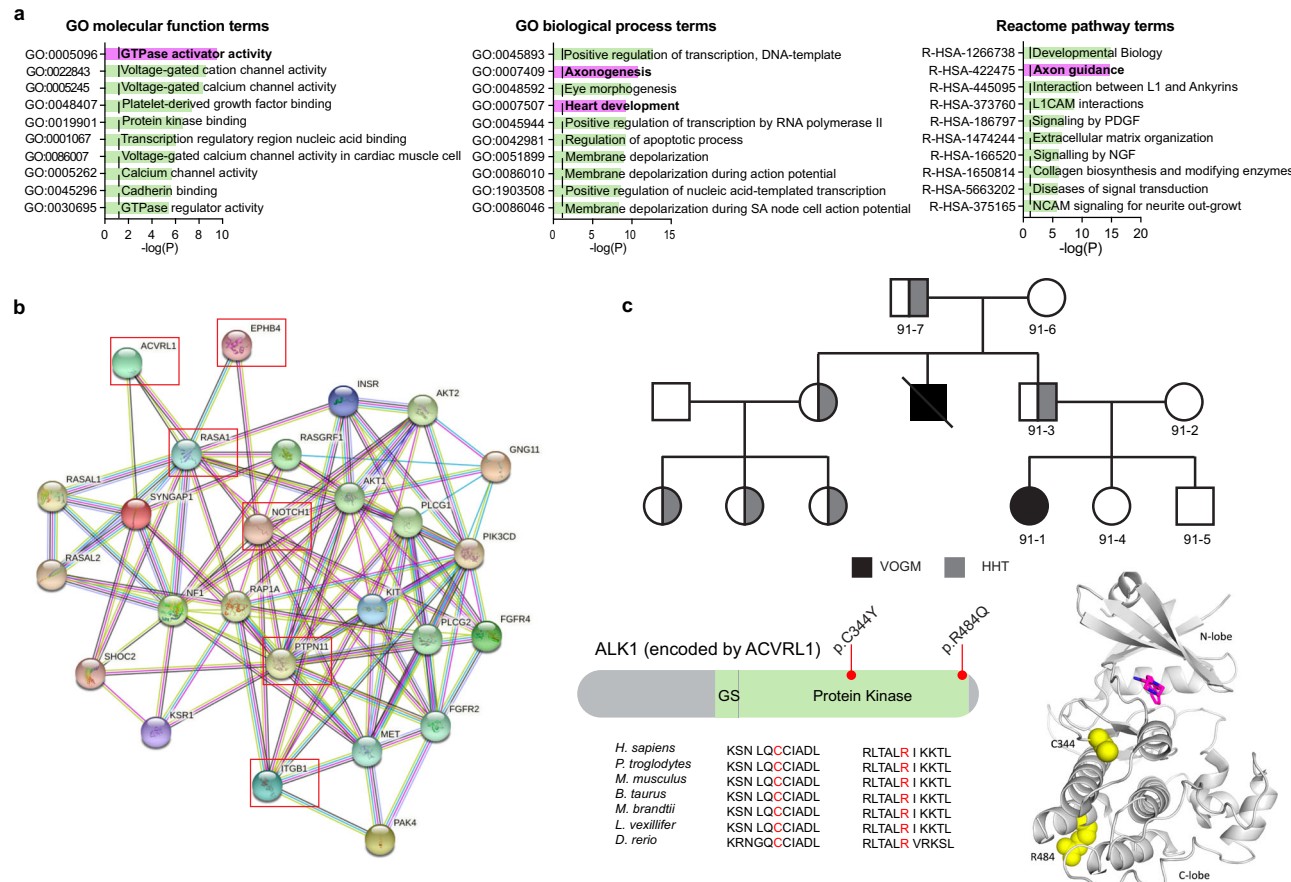

**Fig. 3 | VOGM-associated variants in *ACVRL1* and other Mendelian vascular disease genes. a** Top 10 GO molecular function, GO biological process, and Reactome pathway enrichment terms. The y-axis depicts GO term and Reactome pathway ID numbers. The x-axis depicts -log (*p*-value) and the dotted line represents the $\alpha = 0.05$ significance threshold (one-sided Fisher's exact test, Bonferroni multiple-testing adjusted). The GO term and Reactome pathway term name overline their respective bars. **b** Interactome of mutated Ras signaling genes in VOGM. Genes with damaging Ras signaling variants (WP4223 in Wiki pathway analysis) and *EPHB4*, *NOTCH1*, *ACVRL1*, and *ITGB1* (27 genes in total) were inputted into String (https://string-db.org/cgi/about.pl) and mapped onto a single STRING interactome

that includes a higher than expected number of interactions (PPI enrichment *p*-value < $1.0 \times 10^{-16}$, one-sided hypergeometric test, Bonferroni multiple-testing adjusted). All 6 VOGM risk genes (highlighted by red box) contribute significantly to the PPI enrichment of this network. **c** Multi-generational VOGM family in KVOGM-91. Two transmitted D-mis variants (p. Cys344Tyr and p. Arg484Gln) are located in the kinase domain of ALK1 (*ACVRL1*) shown in ribbon structure (PDB ID: 3MY0)[157]. Cys344 and Arg484 are indicated with yellow spheres. A small molecule inhibitor (purple) is bound in the catalytic cleft. HHT, hereditary hemorrhagic telangiectasia.

Ontology (GO), Reactome, and Wiki pathway analyses (see Methods) on genes of pLI ≥0.9 harboring damaging de novo and/or rare (MAF ≤ $2 \times 10^{-5}$) damaging transmitted variants (see Supplementary Table 8 and Methods). Among the top significantly enriched terms (Fig. 3a and Supplementary Fig. 7) was Reactome pathway term REAC:R-hsa-422475 associated with axon guidance ($p = 2.17 \times 10^{-15}$), a process critical for vascular patterning and regulated by Ephrin-Eph receptor signaling[1]. Other GO biological process and molecular function terms enriched in VOGM probands included those associated with heart development (GO:0007507, $p = 5.19 \times 10^{-10}$) and Ras GTPase activator activity (GO:0005096, $p = 3.35 \times 10^{-10}$) (Fig. 3a).

46 LoF and 72 D-mis damaging variants in total were identified in genes associated with axon guidance terms REAC:R-hsa-422475 (see above) or related axon guidance term GO:0007411 ($p = 2.16 \times 10^{-8}$), including five DNVs (Supplementary Table 9). Besides *RASA1* and *EPHB4*, several other high-pLI genes essential to cerebrovascular development contained multiple rare, damaging de novo or transmitted variants, including *ACVRL1*, *NOTCH1*, *PTPN11*, and *ITGB1* (Supplementary Tables 10–11). These genes also contributed to enrichment signals identified in both the heart development term GO:0007507 and/or the Ras GTPase activator activity term GO:0005096 (Fig. 3a) and mapped onto a robust STRING[71] protein-protein interactome (PPI

enrichment $p < 1.0 \times 10^{-16}$) with other Ras signaling genes harboring rare, damaging variants (Supplementary Table 12 and Fig. 3b). We, therefore, examined the probands harboring these variants in greater detail.

KVOGM73-1 harbored a novel, de novo p.Gly616Cys variant in *NOTCH1* (Supplementary Table 10 and Supplementary Fig. 4), encoding a cell surface receptor for Jagged-1 (*JAG1*) and other Notch family ligands[72]. *NOTCH1* variants are associated with congenital heart defects and other vascular anomalies in AD aortic valve disease type 1 (OMIM# 109730)[73] and AD Adams-Oliver syndrome type 5 (OMIM# 616028)[74]. Although KVOGM73-1 exhibited no structural cardiac pathology, he had moyamoya disease (MMD), in common with *JAG1*-mutant patients with AD Alagille syndrome type 1[75] (OMIM# 118450). Notch proteins are characterized by N-terminal epidermal growth factor (EGF)-like repeats followed by LNR domains which form a complex with ligands to prevent signaling. p.Gly616Cys impacts a conserved NOTCH1 residue required for proper folding of the EGF-like domain (Supplementary Fig. 4)[76]. Unrelated proband KVOGM83-1 contained another rare, damaging, unphased variant in *NOTCH1* (p. Asp1064Asn) at the surface-exposed residue of the 28th EGF repeat, part of the Abruptex region in fly NOTCH and potentially implicated in cis-inhibition of NOTCH1 (Supplementary Fig. 4)[77]. To our knowledge, this is the first

report of co-existing VOGM and MMD, as well as the first report of a de novo *NOTCH1* variant in a VOGM patient.

Proband KVOGM23-1 harbored a de novo p.Tyr63Cys variant in *PTPN11*, encoding the non-receptor tyrosine phosphatase SHP2 (Supplementary Table 10 and Supplementary Fig. 4). *PTPN11* variants, including the identical p.Tyr63Cys variant, have been reported in AD Noonan syndrome type 1 (OMIM# 163950), which features a broad spectrum of congenital heart defects and other systemic vascular lesions[78,79]. KVOGM23-1 exhibited no Noonan syndrome-like features. Interestingly, p.Tyr63Cys impacts a conserved residue of SHP2's N-terminal SH2 domain (Supplementary Fig. 4). Similar to other N-SH2 domain 'blocking loop' variants, p.Tyr63Cys disrupts SHP2 auto-inhibition and causes constitutive Ras signaling activation[80,81]. The paralogous p.Asp61Gly variant in SHP1 potentiates Ras/ERK/MAPK signaling by interfering with the phosphorylation of GAB1, an adapter protein recruiting p120 RasGAP to receptor tyrosine kinases[80,82,83]. To our knowledge, this is the first report of a de novo *PTPN11* variant in a VOGM patient.

Proband KVOGM-91 harbored a rare, damaging variant in *ACVRL1* (p.Cys344Tyr) (Fig. 3c and Supplementary Fig. 4). Variants in *ACVRL1* encoding ALK1, a type I cell-surface TGF-beta superfamily receptor serine/threonine kinase[84], have been implicated in AD hemorrhagic telangiectasia (HHT) type 2 (Rendu-Osler-Weber syndrome 2; OMIM# 600376). Whereas the proband had no features of HHT2, further inquiry revealed that his uncle died at 2 weeks of age from VOGM-related high-output heart failure. Moreover, the proband's father, grandfather, and multiple paternal cousins had stigmata of HHT2, including epistaxis, visceral AVMs (lung and liver), and cutaneous vascular lesions. *ACVRL1* p.Cys344Tyr segregated with VOGM and HHT2 vascular phenotypes in all affected family members available for sequencing. p.Cys344Tyr, impacting a highly conserved residue in the kinase domain (Fig. 3c), has been identified in another HHT2 family (without history of VOGM)[85] and decreases ALK1 cell surface expression[86]. In further support of the pathogenicity of *ACVRL1* variant in VOGM, we found that *Acvrl1a/b* depletion in the *Tg(kdr:gfp)$^{zn1}$* reporter fish line (see Methods) resulted in VOGM-like massive dilation of the venous primordial hindbrain channel and posterior connecting segment, similar to that seen in Ephb4a/b-deficient fish (Fig. 4b and Supplementary Fig. 8). These phenotypes were rescued by mRNA co-injection of wild-type human ACVRL1 but not of VOGM-mutant human ACVRL1 Cys344Tyr (Fig. 4c, d and Supplementary Fig. 9).

Unrelated proband KVOGM-100 contained another rare, damaging variant in *ACVRL1* (p.Arg484Gln) (Fig. 3c and Supplementary Table 10). KVOGM-100 had features of HHT2, including recurrent nosebleeds and pulmonary arterial hypertension (Supplementary Table 11). The proband's father carried the *ACVRL1* variant and had telangiectasias on his tongue, lips, and lower extremities. Variant p.Arg484Gln impacts a conserved surface-exposed residue located in helix αI of the kinase C-lobe, and encodes a catalytically inactive ALK1 mutant[87]. Variants in this region are associated with HHT2, increased incidence of pulmonary arterial hypertension[88–90], and other ALK family-associated diseases such as brachydactyly type A2 (ALK6)[91,92] and Loeys-Dietz syndrome (ALK5)[93,94]. *ACVRL1* p.Arg484Gln has also been identified in HHT2[95] and in childhood-onset pulmonary arterial hypertension[96]. p.Arg484Gln. ALK1 interacts with p120 RasGAP via the Dok-1 adapter protein[97], implicating ALK1-DOK1-p120 RasGAP in crosstalk between the TGF-beta and Ras/ERK/MAPK signaling pathways.

KVOGM89-1 and KVOGM105-1 exomes respectively contained the rare, damaging, variants p.Ser785fs (unphased) and c.2331-2 A>G (transmitted) in *ITGB1* (Integrin subunit beta-1), whose protein product is essential for endothelial cell adhesion, migration, and survival during angiogenesis[98] (Supplementary Fig. 4 and Supplementary Table 10). Interestingly, both patients had intracranial hemorrhage, progressive macrocephaly, and shunt-dependent hydrocephalus. KVOGM105-1 and his sister (unavailable for sequencing) presented with impressive nevus simplex lesions, a type of capillary malformation. p.Ser785fs and c.2331-2 A>G variants are predicted to lead to nonsense-mediated RNA decay. Together, these results suggest that rare, damaging de novo or transmitted variants in other mutation-intolerant genes implicated in other vascular Mendelian syndromes are pathogenic contributors to VOGM.

## VOGM-associated genes are enriched in developing cerebrovascular endothelial cells

To gain insight into the developmental periods, cell types, and molecular pathways involved in VOGM biology, we performed a spatio-temporal consensus Weighted Gene Co-expression Network Analysis (WGCNA)[99], which leverages a large bulk RNAseq data set encompassing multiple human brain regions across development and into early childhood[100]. We constructed 88 modules characterized by genes that share highly similar expression patterns during brain development across different cortical regions and therefore likely to be involved in similar functions[99]. Each module was assessed for relative enrichment of high-confidence and probable (p) VOGM genes, brain AVM genes, AVM+VOGM genes together (since each is a high-flow arterio-venous communication disorder), cerebral cavernous malformation (CCM) genes, moyamoya disease (MMD genes, and human height genes (as a negative control)), using multivariate logistic regression (see Supplementary Table 13 and Methods for description of gene lists).

VOGM genes converged in several modules in the fetal human cortex. The most significant of these was the post-conceptional week (PCW) 37 "Midnight Blue" module ($p = 2.13 \times 10^{-6}$; Fig. 5). The "Black" module was also enriched with VOGM genes (Fig. 5a). The "Black" module exhibited peak gene expression early in development at PCW 9-17 (Fig. 5b). In contrast, the only module to show enrichment for VOGM genes alone, "Light Cyan", is expressed later in neurodevelopment, at 10–12 months postnatally. (Fig. 5b). Notably, genes expressed in the PCW 37 "Midnight Blue" module were enriched for those involved in Focal Adhesion-PI3K-Akt-mTOR Signaling Pathway (WP:3932; $p = 6.73 \times 10^{-10}$, 6.7-fold enrichment), including Positive Regulation of Vascular Development (GO: 1904018; enrichment 11.4-fold; $p = 7.95 \times 10^{-8}$) and Regulation of Cell Migration (GO: 0030334; enrichment 4.4-fold; $p = 5.94 \times 10^{-6}$; Fig. 5c). Human height genes were unenriched in any of the defined modules.

We next studied VOGM gene expression using two different sc-RNAseq transcriptomic atlases. The first set comprised of 154,938 total cells across 42 cell types from 20 brain regions spanning different developmental stages[101]. VOGM genes ($p = 6.43 \times 10^{-6}$) and pVOGM genes ($p = 1.25 \times 10^{-5}$) showed specific enrichment in the endothelial cell (EC) subcluster. VOGM+AVM genes were also enriched in ECs, but the contributors to this signal were VOGM-specific genes, as AVM genes alone were not significantly enriched. In contrast, CCM and MMD genes were not enriched in Ecs. VOGM genes were also enriched in other cell types to lesser degrees than in EC subtypes. Examples include the excitatory embryonic neuron (cluster "ExN6b", $p = 4.75 \times 10^{-4}$) and the inhibitory neurons (cluster "InN4b", $p = 3.18 \times 10^{-4}$) (Fig. 5d). Control human height genes were unenriched in any of the 42 cell subtypes.

The second sc-RNAseq transcriptomic atlas included 181,388 total cells from normal and diseased human cerebrovasculature, including both endothelial and perivascular cell subtypes[46]. Corroborating results from the STAB dataset, VOGM genes showed specific enrichment in ECs ($p = 6.43 \times 10^{-6}$). VOGM, AVM, and AVM+VOGM genes were enriched in arterial and venous EC subtypes, but not in perivascular subtypes such as smooth muscle cells, fibroblasts, fibromyocytes, pericytes, or other perivascular cell types (Fig. 5e). In contrast, CCM and MMD genes were unenriched in arterial and venous EC subtypes. Human height genes were not enriched in any of the 18 cell subtypes.

We again constructed modules from the cerebrovascular dataset comprising co-expressed genes. These were then assessed for the

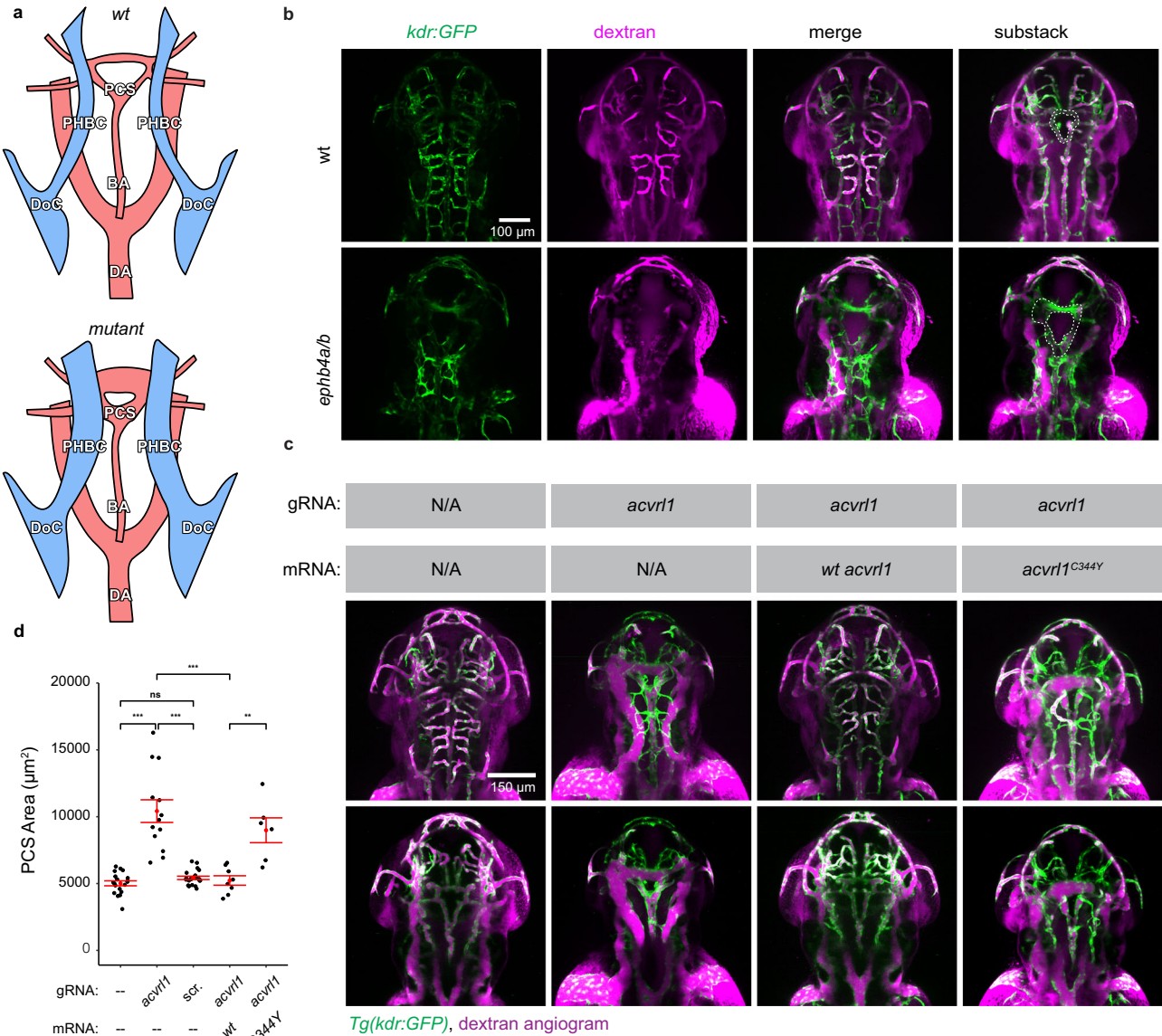

**Fig. 4 | Depleting VOGM candidate genes in zebrafish lead to aneurysm-like phenotypes. a** Schematic showing layout of cranial arteries (red) and veins (blue) in 48 hpf zebrafish *wt* and mutant embryos. We attempted to identify phenotypes in any of the following vessels: primordial hindbrain channel (PHBC), basilar artery (BA), posterior communicating segment (PCS), and dorsal aorta (DA). Also pictured: duct of Cuvier (DoC). Generally, we observed enlarged PHBC and PCS vessels. **b** Cranial vasculature indicated by Tg(kdr:gfp) and dextran microangiography of 48 hpf wt and ephb4a/b-depleted zebrafish embryo. Scale bar: 100 μm. Cranial vasculature is indicated by Tg(kdr:gfp), dextran microangiogram, and merged channels of the wt and ephb4a/b-deleted embryo. Ventrally restricted confocal substacks emphasizing PCS (dashed lines) are shown. Ventrally restricted confocal substack which emphasizing PCS (dashed lines) shows enlarged PCS in ephb4a/b-depleted zebrafish. Altogether, 46 independent larvae were imaged from 15 independent experiments. **c** Cranial vasculature indicated by Tg(kdr:gfp) and dextran microangiography of 48 hpf embryos following acvrl1 gRNA targeting, rescue, and false rescue with C344Y variant. Scale bar: 150 μm. Cranial vasculature is indicated by Tg(kdr:gfp) and dextran microangiogram (merged). Ventrally restricted confocal substacks emphasizing PCS (lower row) are shown. This confocal substack shows enlarged PCS in targeted embryos which can be rescued with wt mRNA but not C344Y mRNA. Note that PHBC is also enlarged in targeted embryos. **d** Quantification of PCS results (two-sided Wilcoxon signed-rank test, not adjusted). we also injected scrambled acvrl1 gRNA as a further negative control. *** ≤ 0.001, ** ≤ 0.01, * ≤ 0.05. n = 66 independent larvae taken from 14 independent experiments. Error bars are ± standard error of the mean.

enrichment of VOGM and other disease genes (see Methods). Of the 25 defined modules, VOGM genes were most highly enriched in Modules 11 ($p = 3.23 \times 10^{-4}$) and 12 ($p = 8.22 \times 10^{-5}$), overlapping with AVM genes (Fig. 5f). In contrast, CCM and MMD genes were most enriched in Module 6 defined by mitochondrial processes. Human height genes were not significantly enriched in any of the modules (Fig. 5f). Module 11 genes were most highly enriched for pathways related to VEGFA-VEGFR2 signaling (WP3888; 1.97-fold enrichment; $p = 4.51 \times 10^{-5}$), vascular transport (GO:0010232; 5.0-fold enrichment; $p = 7.04 \times 10^{-8}$), and GTPase activator activity (GO:0005096; 2.7-fold enrichment; $p = 3.89 \times 10^{-8}$) (Fig. 5g and Supplementary Fig. 10). Module 12 was notably

enriched for pathways related to Ras/ERK/MAPK (WP4223; 2.3-fold enrichment; $p = 3.89 \times 10^{-3}$) and inflammation-regulated EGFR signaling (WP4483; 6.5-fold enrichment; $p = 2.05 \times 10^{-3}$) (Fig. 5g and Supplementary Fig. 10). Taken together, these results suggest developing cerebral endothelial cells are an important spatio-temporal locus impacted by VOGM-associated gene variants.

## Phenotypic validation of a VOGM-specific EPHB4 kinase domain missense variant in mice

Mice constitutively deficient in EPHB4 or its ligand, Ephrin B2, expire at E10.5 of gestation because vascular plexuses that arise

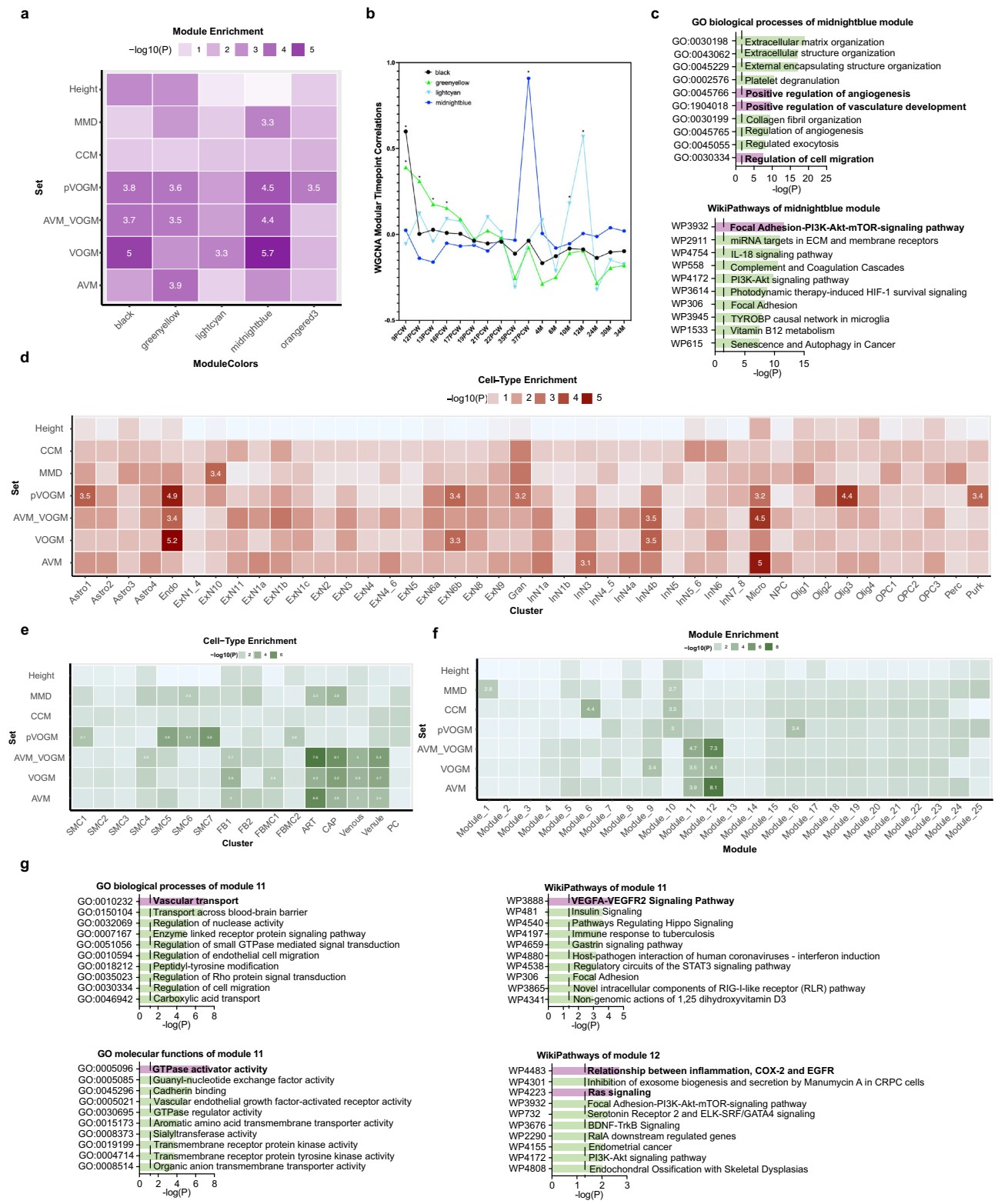

through vasculogenesis are not remodeled by angiogenesis into hierarchical arterial-capillary-venous networks[3,102,103]. However, whether EPHB4 kinase activity is required for normal blood vascular development remains unknown. Based on our human genetic (Fig. 1) and integrative genomic results (Fig. 5), we hypothesized that genetic inactivation of a kinase-dependent, EPHB4-regulated RASA1 signaling mechanism in endothelial cells disrupts VEGF-associated developmental angiogenesis in vivo. To test this hypothesis and gain insight into the mechanism of the identified

VOGM-associated EPHB4 kinase domain missense variants, we generated *EphB4* mutant mice carrying a knock-in allele ortholo-gous to the *EPHB4* p.Phe867Leu variant identified in patient KVOGM18-1 (Table 2 and Methods).

Heterozygous *Ephb4*[+/F867L] mice were viable, fertile, and exhibited no obvious vascular developmental or other defects. However, homozygous *Ephb4*[F867L/F867L] pups were not identified among the pro-geny of heterozygous *Ephb4*[+/F867L] parents, suggesting that the EPHB4 p.Phe867Leu variant is lethal in homozygous form (Fig. 6a).

**Fig. 5 | VOGM genes converge in a VEGFR-Ras signaling network in fetal cerebral endothelial cells. a** Enrichment in VOGM gene modules of the fetal human cortex, compared to other disease genes. Numbers displayed exceed the Bonferroni-corrected statistical significance threshold tested by one-sided Fisher's exact test and are -$\log_{10}$(*p*-value). Height: human height; MMD: moyamoya disease; CCM: cavernous malformation; AVM_VOGM: arteriovenous malformation and vein of Galen aneurysmal malformation; VOGM: high-confidence vein of Galen aneurysmal malformation gene set; pVOGM: probable VOGM gene set; AVM: arteriovenous malformation (see Methods for gene set details). **b** Temporal dynamics of modules enriched with VOGM genes. Peak expression of "Midnight Blue" module is at post-conception week (PCW) 37. Both the "Black" and "Green-yellow" modules exhibited peak gene expression early in development at PCW 9-17. "Light Cyan" module is expressed much later at postnatal age 10–12 months. **c** Gene Ontology (GO) biological processes and WikiPathways of midnight blue module converge on Focal Adhesion-PI3K-Akt-mTOR Signaling Pathway, Positive Regulation of Vascular Development, and Regulation of Cell Migration (one-sided Fisher's exact test,

Bonferroni multiple-testing adjusted). The significance threshold is denoted by the vertical dashed line. Top enriched terms have bolded text and purple bars. **d** Cell-type enrichment of VOGM and other disease genes in the fetal human cortex. Numbers displayed exceed the Bonferroni-corrected statistical significance threshold tested by one-sided Fisher's exact test and are -$\log_{10}$(*p*-value). Different cell types are noted on the x-axis, see text for details. **e** Cell-type enrichment of VOGM genes in the developing human cerebrovasculature. Numbers displayed exceed the Bonferroni-corrected statistical significance threshold tested by one-sided Fisher's exact test and are -$\log_{10}$(*p*-value). Different cell types are noted on the x-axis (see text for details). **f** Enrichment of disease genes in fetal human cortex modules. Numbers displayed exceed the Bonferroni-corrected statistical significance threshold tested by one-sided Fisher's exact test and are -$\log_{10}$(*p*-value). **g** GO molecular function and WikiPathways analyses of enriched module 11 and 12 (one-sided Fisher's exact test, Bonferroni multiple-testing adjusted). The significance threshold is denoted by the vertical dashed line. Enriched terms of interest have bolded text and purple bars.

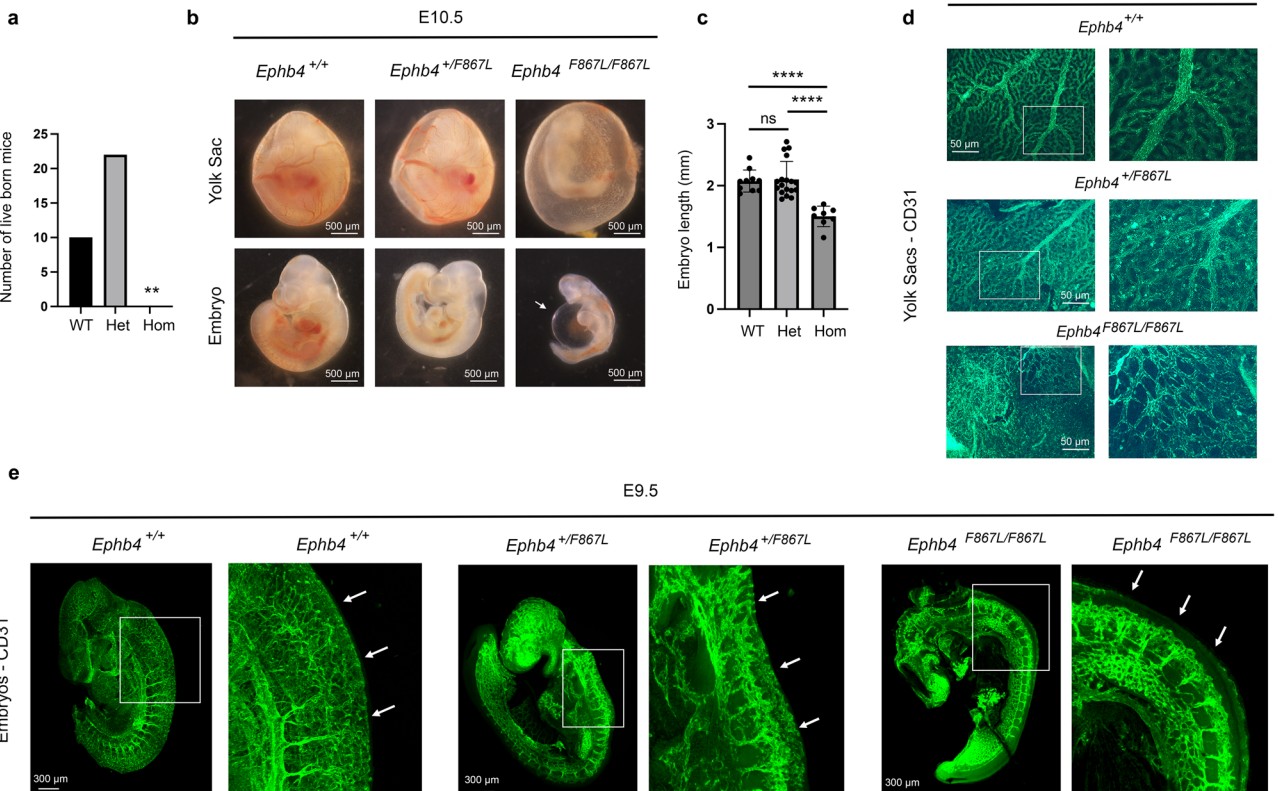

**Fig. 6 | VOGM-specific EPHB4 variant disrupts developmental angiogenesis.** *Ephb4*[+/F867L] mice were intercrossed. **a** Number of live born *Ephb4*[+/+] wild-type (WT), *Ephb4*[+/F867L] heterozygote (Het), and *Ephb4*[F867L/F867L] homozygote (Hom) mice. **p* = 0.0046; Chi-squared test (two-sided). **b** Images of littermate embryos of the indicated genotypes at E10.5. Note abnormal vascularization of the yolk sac, reduced size and distended pericardial sac (arrow) of homozygous *Ephb4*[F867L/F867L] embryos. **c** Mean +/- 1 SEM of embryo length at E10.5 (WT, *n* = 10; Het, *n* = 18; Hom, *n* = 8). ****$P < 0.0001$; ns, not significant; Student's two-sample two-sided *t*-test. **d** E10.5 yolk sacs were stained with anti-CD31 antibodies to identify vasculature.

Images at right are higher magnifications of the boxed areas of images at left. Note hierarchical vascular networks in yolk sacs of wild-type and heterozygous embryos and primitive vascular plexuses in yolk sacs of homozygous *Ephb4*[F867L/F867L] embryos that present as classical honeycomb-like structures at higher magnification. **e** CD31 antibody staining of E9.5 embryos of the indicated genotypes. For each embryo, the right image is a higher magnification of the boxed area in the left image. Arrows show trunk angiogenesis toward the midline in wild-type and heterozygous embryos but not homozygous *Ephb4*[F867L/F867L] embryos. Images in (**d**, **e**) are representative of at least 2 embryos of each genotype.

Examination of homozygous *Ephb4*[F867L/F867L] embryos at E10.5 confirmed a smaller size and vascular defects similar to those reported in constitutive EPHB4- and Ephrin B2-deficient E10.5 embryos (Fig. 6b, c). Blood flow was often absent in E10.5 *Ephb4*[F867L/F867L] embryos and the pericardial sac was grossly distended (Fig. 6b). The yolk sacs of littermate E10.5 *Ephb4*[+/+] and *Ephb4*[+/F867L] embryos showed normal hierarchical vascular networks supplying blood to the developing embryo (Fig. 6b and d). In contrast, the yolk sacs of *Ephb4*[F867L/F867L]

embryos exhibited honeycomb-like vascular patterning typical of primitive vascular networks (Fig. 6b and d). At E10.5, some *Ephb4*[F867L/F867L] embryos showed reduced cellularity (not shown). Therefore, we examined E9.5 *Ephb4*[F867L/F867L] embryos to characterize vascular defects in the embryo proper. E9.5 *Ephb4*[F867L/F867L] embryos exhibited impaired developmental angiogenesis among intersomitic vessels that failed to extend branches toward the midline as observed in *Ephb4*[+/+] and *Ephb4*[+/F867L] embryos (Fig. 6e). The same phenotype was previously

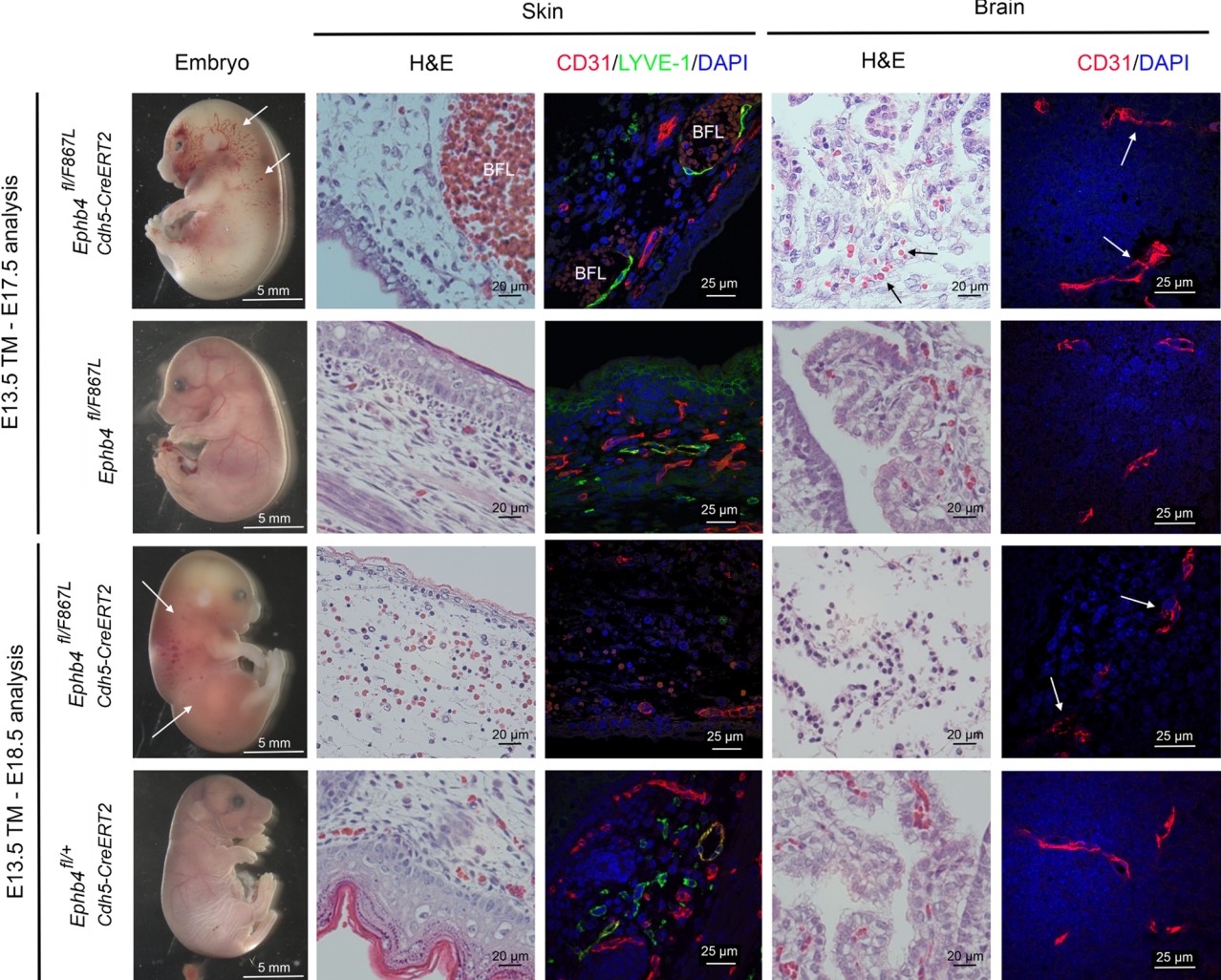

**Fig. 7 | Vascular-specific expression of EPHB4 F867L alone during mid to late gestation results in blood and lymphatic vascular abnormalities.** Embryos of the indicated genotypes were administered tamoxifen (TM) at E13.5 and harvested at E17.5 or E18.5. At left are images of whole embryos ($n = 3$ each genotype). Note evidence of blood-filled lymphatics (BFL) in skin of E17.5 *Ephb4^fl/F867L Cdh5-CreERT2* embryos (arrows) but not in control littermate *Ephb4^fl/F867L* embryos. Staining of skin sections with H&E and anti-CD31, and anti-LYVE-1 antibodies revealed dilated blood-filled lymphatics in *Ephb4^fl/F867L Cdh5-CreERT2* embryos. Note generalized hemorrhage in skin of E18.5 *Ephb4^fl/F867L Cdh5-CreERT2* embryos (arrows) but not in control littermate *Ephb4^fl/+ Cdh5-CreERT2* embryos, confirmed by staining of tissue sections with H&E. Staining of E18.5 tissue sections with anti-CD31 anti-LYVE-1 antibodies revealed near complete absence of lymphatic vessels in skin of E18.5 *Ephb4^fl/F867L Cdh5-CreERT2* embryos compared to E17.5 and E18.5 controls. Images of H&E-stained brain sections are of the cerebellar/choroid plexus region. Note evidence of hemorrhage (extravascular erythrocytes) in E17.5 *Ephb4^fl/F867L Cdh5-CreERT2* embryos (arrows) but not in controls. The E18.5 brain of *Ephb4^fl/F867L Cdh5-CreERT2* embryos shows extensive loss of cellularity and total absence of erythrocytes. CD31 staining of brain sections shows disorganized vascular structures in *Ephb4^fl/F867L Cdh5-CreERT2* embryos compared to controls in the region of the choroid plexus and cerebellum.

noted in constitutive EPHB4- and Ephrin B2-deficient embryos at E9.5[103].

We have shown previously that EPHB4 and RASA1 promote survival of endothelial cells during developmental angiogenesis[67,104]. Disruption of *Ephb4* or *Rasa1* genes in endothelial cells in mid-to-late gestation (when vasculogenesis has ceased and angiogenesis predominates) impairs their ability to export the extracellular matrix protein, collagen IV, a major constituent of vascular basement membranes. Consequently, endothelial cells undergo apoptosis secondary to an unfolded protein response triggered by collagen IV trapped within the endoplasmic reticulum, and/or secondary to poor adhesion to defective basement membrane. Therefore, to further understand the mechanism of vascular dysfunction in EPHB4 Phe867Leu embryos, we generated *Ephb4^fl/F867L* embryos carrying the *Cdh5-CreERT2* endothelial-specific inducible cre driver (see Methods). At E13.5 of development, tamoxifen was administered to embryos to disrupt the *Ephb4* floxed allele, resulting in endothelial-specific expression of

EPHB4 Phe867Leu alone. At E17.5, *Ephb4^fl/F867L Cdh5-CreERT2* embryos showed evidence of blood-filled lymphatic vessels in skin, confirmed by H&E staining and immunostaining of tissue sections with antibody against the lymphatic marker, LYVE-1 (Fig. 7). The presence of blood-filled lymphatics is consistent with the known requirement of EPHB4 for development of lymphovenous valves that prevent backflow of venous blood into the lymphatic vascular system[105]. Strikingly, at E18.5 lymphatic vessels were essentially absent from skin, and there was evidence of extensive cutaneous hemorrhage (Fig. 7). We previously reported the same phenotypes in *Ephb4^fl/fl Cdh5-CreERT2* embryos treated with tamoxifen at E13.5[67]. These phenotypes were not evident in control tamoxifen-treated *Ephb4^fl/+ Cdh5-CreERT2* embryo littermates that retain a single wild-type *Ephb4* gene copy after treatment (Fig. 7).

We also examined sections of brain tissue from tamoxifen-treated *Ephb4^fl/F867L Cdh5-CreERT2* embryos for evidence of vascular defects. At E17.5, blood vascular hemorrhage in the region of the cerebellum and

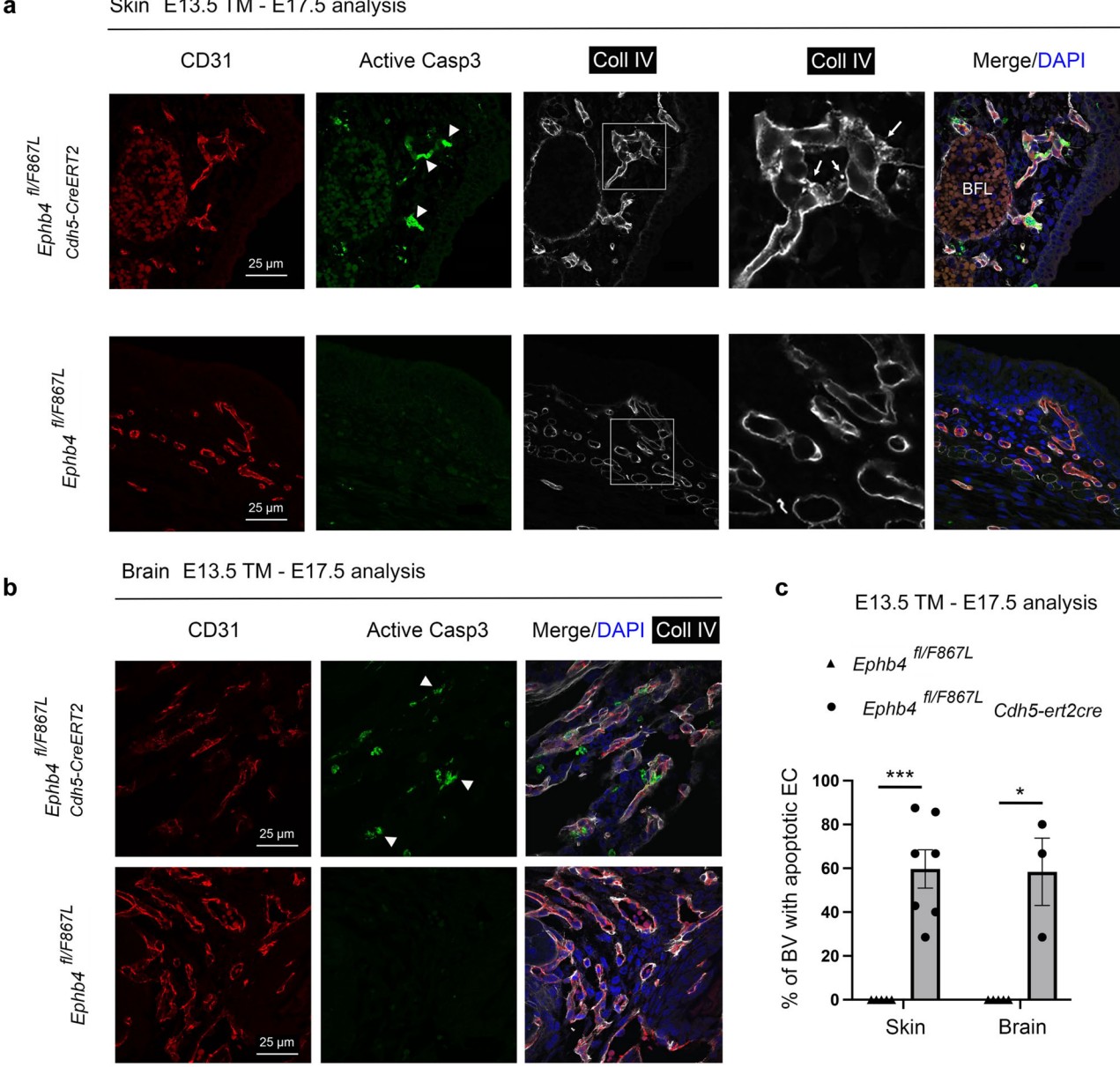

**Fig. 8 | Apoptotic death of skin and brain endothelial cells in embryos induced to express EPHB4 F867L alone within the vasculature during mid-to late gestation.** Embryos of indicated genotypes (*n* = 3) were administered tamoxifen at E13.5 and harvested at E17.5. Sections of skin (**a**) and brain (**b**) were stained with antibodies against CD31, active caspase-3, and collagen IV (Coll IV). Note active caspase-3 positive endothelial cells in blood vessels of skin and brain of *Ephb4*^fl/F867L^ *Cdh5-CreERT2* embryos (arrowheads) but not in littermate *Ephb4*^fl/F867L^ controls.

Note also intracellular accumulation of collagen IV in skin vascular endothelial cells of *Ephb4*^fl/F867L^ *Cdh5-CreERT2* embryos (arrows) but not in controls. **c** Mean +/-1 SEM of % of vessels with at least one apoptotic endothelial cell per field of the choroidal plexus/cerebellum region (*Ephb4 fl/F867L* skin, *n* = 5; *Ephb4 fl/F867L Cdh5-ert2cre* skin, *n* = 7; *Ephb4 fl/F867L* brain, *n* = 5; *Ephb4 fl/F867L Cdh5-ert2cre* brain, *n* = 3 independent fields). \*\**p* = 0.0025; \**p* = 0.0179; two-sided Mann–Whitney test.

choroid plexus was observed (Fig. 7). However, at E18.5, neither intravascular nor extravascular erythrocytes were identified in the brain, and brain cellularity was substantially reduced (Fig. 7). At both developmental time points, CD31-positive blood vessels appeared disorganized as compared to control brain blood vessels (Fig. 7). To examine if endothelial cells in *Ephb4*^fl/F867L^ *Cdh5-CreERT2* embryos underwent abnormal apoptosis after tamoxifen treatment, tissue sections from E17.5 embryos were stained with antibodies against active caspase-3. In both skin and the choroidal plexus/cerebellar region of brains, apoptotic vascular endothelial cells were identified (Fig. 8a–c). Furthermore, intracellular accumulation of collagen IV was evident in skin (Fig. 8a). In summary, mid-to-late gestation embryos induced to express only EPHB4 Phe867Leu specifically in endothelial

cells phenocopy embryos induced to lose endothelial expression of EPHB4 or RASA1 during the same period.

## Discussion

The genetic understanding of VOGM has been hindered by its rarity and sporadic nature. Here, we have presented the largest WES cohort of VOGM to date and integrated these WES results with human brain and vasculature transcriptomes to define the cell types, developmental time points, and functional networks impacted by genes associated with VOGM. We also validated candidate genes in zebrafish and mouse models of disease. Collectively, these data shed insight into the cellular and molecular pathogenesis of VOGM, help illuminate the genetic regulation of human arterio-venous development and have

implications for clinical risk assessment of patients and their families and the development of targeted therapeutics.

Our analysis uncovered enrichment of rare, heterozygous, damaging de novo and inherited germline variants in VOGM probands (Supplementary Data 3) in mutation-intolerant (high-pLI) signaling genes highly expressed in the embryonic vasculature[106] and known to regulate cerebrovasculature development[107,108]. These disease-associated genes map onto a robust STRING[71] protein-protein inter-actome (Fig. 3) that includes p120 RasGAP (*RASA1*) and EphB4 (*EPHB4*), which cooperates with RASA1 to regulate Ras signaling in endothelial cells. Consistent with this, gain-of-function variants were identified in the tyrosine phosphatase SHP2 (*PTNPN11*), which binds to and dephosphorylates Ras to increase its association with Raf and activate Ras signaling[109,110]. Moreover, LoF variants in the serine/threonine receptor kinase ALK1 (*ACVRL1*), which facilitates crosstalk between the TGF-beta and Ras signaling pathways by associating with RASA1 via the Dok-1 adapter protein[97], were detected in other probands, including an ultra-rare multi-generational VOGM family. These data suggest that genetic dysregulation of Ras signaling is an important driver of VOGM pathogenesis, consistent with other cerebral vascular anomalies[25,27–29,111–114].

Germline variants in VOGM risk genes *RASA1*, *EPHB4*, *ACVRL1*, *PTNPN11*, and *NOTCH1* have been reported in other Mendelian diseases featuring vascular phenotypes[25,27,110,111,115–117]. *RASA1* and *EPHB4* encode interacting proteins mutated in CM-AVM type 1 and 2, respectively. Our VOGM probands with *ACVRL1* (ALK1) variants, which have been linked with HHT1, are either part of a multi-generational VOGM family with HHT1 or have HHT1 features. Consistent with this pattern is the previous identification of a VOGM patient with a variant in HHT2 gene *ENG*, encoding the ALK1 binding partner Endoglin[28]. Interestingly, the Ras-activating *PTPN11* (SHP2) p.Tyr63Cys variant identified in KVOGM23-1 was previously identified in Noonan syndrome (OMIM# 163950), which features a broad spectrum of congenital heart defects and other systemic vascular lesions[78,79]. Thus, VOGMs may represent phenotypic expansions of CM-AVM, HHT, or other Mendelian vascular syndromes. Nonetheless, none of our *RASA1* or *EPHB4* patients or their family members carried a diagnosis of CM-AVM at the time of DNA sequencing.

Inherited VOGM variants show incomplete penetrance and variable expressivity, with variant carriers often exhibiting cutaneous vascular lesions. Cutaneous vascular lesions are a common hallmark of developmental vascular disorders such as *RASA1*- and *EPHB4*-mutated CM-AVM[25,27], *ENG1*- and *ACVRL1*-mutated HHT[29], and *RASA1*-mutated Parkes Weber syndrome, among others[30,118,119]. As many VOGM families with full clinical data had capillary malformations or other uncommon cutaneous vascular lesions, and as variants identified in probands were found in all family members with these cutaneous lesions, VOGMs and the cutaneous lesions are likely linked to the same variants. These observations highlight the pleiotropy of these genetic variants and the value of WES as a diagnostic adjunct.

Variable expressivity of VOGM and associated features could arise from environmental modifiers[120] in concert with the identified rare variants and/or specific genetic modifiers[43]. However, our findings are most consistent with a two-hit mechanism in which phenotypic expression relies on an inherited germline mutation and a second, post-zygotic (i.e., somatic) mutation in a second allele[121,122]. This mechanism has been shown for other hereditary multifocal vascular malformations, such as CM-AVM1[27,123,124], glomuvenous malformations (OMIM: 138000), cutaneomucosal venous malformation (OMIM: 600195), and cerebral cavernous malformations (OMIM: 116860)[122]. In this context, phenotypic expression depends on the cell types in which somatic variants co-occur and could explain the low penetrance of VOGM arising from transmitted variants. Our data in mice supports this genetic mechanism, as heterozygous *Ephb4*[F867L] mice expressing a

VOGM-specific *EPHB4* kinase-domain missense variant exhibited severe vascular abnormalities, but only when carrying a second variant allele, i.e. *Ephb4*[F867L] or a cre-disrupted *Ephb4* allele. Exome sequencing of ultra-rare human VOGM lesion tissue could test this two-hit hypothesis.

Integration of our WES findings with scRNA-seq datasets of the developing human cerebrovasculature helped define developing endothelial cells as a potential key spatio-temporal locus of VOGM pathophysiology (see Fig. 5). Pathway analysis showed VOGM genes are enriched in growth-factor-regulated, tyrosine receptor kinase-associated signaling, which regulates vasculogenesis, angiogenesis, and arterio-venous specification[1]. Thus, tamoxifen-induced *Ephb4*[fl/F867L] *Cdh5-CreERT2* mice expressing a VOGM-patient muta-tion that disrupts EphB4 kinase activity in fetal endothelial cells exhibited impaired remodeling of primitive vascular plexuses by VEGF-regulated sprouting angiogenesis, a process required for development of hierarchical arterial-capillary-venous networks[67]. VOGM genes are also enriched in pathways that regulate axon pathfinding, which is important for vascular patterning[1]. Elucidating the mechanistic details of how VOGM variants impact vasculogen-esis, including the intercellular communication between develop-ing neural and endothelial cell types, will be important topics of future investigation.

Although rare, damaging DNVs of large effect contribute ~12% of VOGM cases, variants in additional genes likely contribute to disease pathogenesis. Our Monte Carlo simulation based on observed dama-ging DNVs estimated ~66 genes contribute to VOGM by a de novo mechanism. Future WES of 250 or of 1,000 trios should yield respec-tive saturation rates of 20.1% and 70.5%, respectively (Supplementary Fig. 11). The consequence of cis-regulatory elements and more com-plex structural variants remains unknown. The still unclear impact of somatic variants requires additional deep sequencing in matched normal and affected-tissue pairs to enable their characterization. Genomic repeats, transposable elements, and epigenomic changes should also be investigated using more comprehensive long-read sequencing technologies.

Our findings have several clinical implications. The enrichment of damaging variants in the VOGM cohort suggests that variant carrier offspring may be at increased risk for VOGMs as well as for capillary malformations and potentially other AVMs. However, not all variant carriers develop capillary malformations, making the presence of capillary malformations an unreliable clinical marker for transmission risk in affected families. These observations highlight the importance of family history and the potential use of exome sequencing-based screening for risk assessment among family members. Family mem-bers of VOGM patients with positive exome sequencing results might benefit from intracranial imaging with MRI/MRA, especially in the setting of suspicious mucocutaneous lesions.

The finding that MAPK inhibitors rescue *RASA1* and *EPHB4*-mutant embryonic vascular phenotypes in mice[67,104,125] suggest potential tar-geted treatments for VOGM and perhaps other CM-AVM and HHT spectrum lesions[67,104,125]. Trametinib, an orally available inhibitor of MEK kinase activity, is under evaluation in a prospective phase II trial for systemic AVMs (TRAMAV (Trametinib in Arterio-Venous Mal-formations that are refractory to standard care, https://www.clinicaltrialsreg- ister.eu; unique identifier: 2019-003573-26)). In con-trast to systemic vascular lesions, the narrow window of gestational weeks 6–11 during which the primitive choroidal arteries and the MPV are usually present, and therefore during which VOGMs are speculated to develop[10], may pose a challenge to improved early therapeutic strategies for VOGM. Thus, attempted diagnosis with intention to treat must proceed before the safe gestational age threshold for amniocentesis[126]. These data highlight the need for continued genetic research on VOGM, as well as investigation into the cellular and

molecular mechanisms of newly discovered VOGM-associated variants in mammalian model systems.

## Methods

All procedures in this study comply with Yale University's Human Investigation Committee (HIC) and are approved by Yale University's Human Research Protection Program. Written informed consent for genetic studies was obtained from all participants. Written authorization was obtained from a parent or legal guardian for sample collection from all minors in this study. All animal care and handling was performed in accordance with Yale IACUC protocol 2019-20274. All experiments with mice were approved by the Yale University and University of Michigan IACUCs.

### Patient subjects

Inclusion criteria included male or female patients with clearly defined mural or choroidal VOGMs, radiographically confirmed by both a neurosurgeon and neuroradiologist from an angiogram or magnetic resonance angiogram. Family members of included patients were also studied when possible. Given the non-randomized cohort design of this study aimed at identifying genetic etiologies associated with VOGM, it is important to note that the demographic distribution, including the sex ratio, may not be representative of the general population. Controls consisted of 1,798 unaffected siblings of autism cases and unaffected parents from the Simons Foundation Autism Research Initiative Simplex Collection (SFARI)[49,50]. Only the unaffected siblings and parents, as designated by SFARI, were included in the analysis and served as controls for this study. Permission to access the genomic data in the SFARI or the National Institute of Mental Health Data Repository was obtained. The SFARI provided written informed consent for all participants.

### Exome sequencing and variant calling

Exome capture was performed on genomic DNA samples derived from saliva or blood using Roche SeqCap EZ MedExome Target Enrichment kit or IDT xGen target capture kit followed by 99 base paired-end sequencing on the Illumina sequencing platform. Sequence reads were aligned to the human reference genome GRCh37/hg19 using BWA-MEM[127] and further processed to call variants following the GATK Best Practices workflow[52] and Freebayes[54]. Variants were annotated with ANNOVAR[128], gnomAD (v.2.1.1)[129], and BRAVO[55] databases. We used MetaSVM and MPC algorithms to predict the deleteriousness of missense variants (D-Mis, defined as MetaSVM-deleterious or MPC-score $\geq$ 2)[57,130]. Inferred LoF variants consist of stop-gain, stop-loss, frameshift insertions/deletions, canonical splice sites, and start-loss. LoF and D-Mis variants were considered 'damaging.' DNVs were called using TrioDeNovo[56]. Candidate DNVs were further filtered based on the following criteria: (1) exonic or splice-site variants; (2) a minimum read depth (DP) of 10 in the proband and both parents; (3) minimum proband alternative read depth of 5; (4) proband alternative allele ratio $\geq$ 28% if having <10 alternative reads or $\geq$20% if having $\geq$10 alternative reads; (5) alternative allele ratio in both parents $\leq$ 3.5%; (6) global MAF $\leq$4 x $10^{-4}$ in the Exome Aggregation Consortium database. For recessive variant analysis, we filtered for rare (MAF $\leq$ 1 × $10^{-3}$ in BRAVO and in-cohort MAF $\leq$1 × $10^{-2}$) homozygous and compound heterozygous variants that exhibited high-quality sequence reads (pass GATK variant quality score recalibration, $\geq$ 8 total reads for proband, genotype quality (GQ) score $\geq$ 20). Only LoF, D-Mis, and non-frameshift indels were considered potentially deleterious to the disease. Only homozygous variants were analyzed for probands with unavailable parental WES data. See Supplementary Table 14 for damaging recessive variants identified. For rare transmitted dominant variants, only LoF and D-Mis were included. They were filtered using the following criteria: (1) pass GATK variant quality score recalibration, (2) MAF $\leq$ 5 × $10^{-5}$ in Bravo and in-cohort MAF $\leq$ 5 × $10^{-3}$;

(3) a minimum DP of 8 in the proband, and (4) GQ score $\geq$ 20. After filtering each type of variant as described above, false-positive calls were removed by in silico visualization. Candidate variants were confirmed by PCR amplification followed by Sanger sequencing (primer sequences available on request).

### Kinship analysis

The relationship between proband and parents was estimated using the pairwise identity-by-descent (IBD) calculation in PLINK[131]. IBD sharing between the proband and parents in all trios was between 45% and 55%. For pairs sharing $\geq$ 80% of rare variants, the sample with greater sequence coverage was retained in the analysis, and the other discarded.

### Principal component analysis

Subject ethnicity was determined by EIGENSTRAT[132] software to analyze tag SNPs in cases, controls, and HapMap subjects, as described[39].

### De novo expectation model

As case trios were captured by two different reagents (MedExome and IDT), we took the union of all bases covered by different capture reagents. We generated a Browser Extensible Data file representing a unified capture for all trios. We used bedtools (v.2.27.1) to extract sequences from the Browser Extensible Data file. We then applied a sequence context-based method to calculate the probability of observing a DNV for each base in the coding region, adjusting for sequencing depth in each gene[133]. Briefly, for each base in the exome, the probability of observing every trinucleotide mutating to other trinucleotides was determined. ANNOVAR (v2015Mar22) was used to annotate the consequence of each possible substitution. RefSeq was used to annotate variants (based on the file 'hg19_refGene.txt' provided by ANNOVAR). For each gene, the coding consequence of each potential substitution was summed for each functional class (synonymous, missense, canonical splice site, frameshift insertions/deletions, stop-gain, stop-loss, and start-loss) to determine gene-specific mutation probabilities[133]. The probability of a frameshift variant was determined by multiplying the probability of a stop-gain variant by 1.25[133]. The model does not account for in-frame insertions or deletions. To align with ANNOVAR annotations, analysis was limited to variants located in exonic or canonical splice site regions and were not annotated as 'unknown' by ANNOVAR. We identified potential coding variants and generated gene-specific mutation probabilities for 19,347 unique genes following the inclusion criteria. Due to the difference between exome capture kits and DNA sequencing platforms, and to variable sequencing coverage between case and control cohorts, separate de novo probability tables were generated for cases and 1,798 autism control trios.

### De novo enrichment analysis

The burden of DNVs in the case and control trios was determined using the denovolyzeR package[39,134]. Briefly, the expected number of DNVs in the case and control cohorts across each functional class was calculated by taking the sum of each functional class-specific probability multiplied by the number of probands in the study 2× (diploid genomes). Then, the expected number of DNVs across functional classes was compared to the observed number in each study using a one-sided Poisson test. To examine whether any individual gene contains a greater number of protein-altering DNVs than expected, the expected number of protein-altering DNVs was calculated from the corresponding probability, adjusting for cohort size. A one-sided Poisson test was then used to compare the observed DNVs for each gene versus expected. As separate tests were performed for protein-altering, protein-damaging, and LoF DNVs, the Bonferroni multiple-testing threshold is, therefore, equal to 8.6 × $10^{-7}$ (= 0.05 / (3 tests × 19,347 genes)).

## Estimation of the expected number of rare, transmitted variants

We implemented a one-sided binomial test to quantify enrichment of rare damaging or LoF heterozygous variants in each gene by comparing observed and expected counts estimated by mutability[39]. Because the number of rare damaging or LoF heterozygous variants in a gene was inversely correlated with the constraint score (i.e., pLI score) obtained from the gnomAD database, we stratified genes into five subsets by pLI quartiles to control for the potential confounding effect: (1) the first quartile with pLI $<6.4 \times 10^{-8}$; (2) the second quartile with pLI score $6.4 \times 10^{-8}$–$1.9 \times 10^{-3}$; (3) the third quartile with pLI $1.9 \times 10^{-3}$ –0.48; (4) the fourth quartile with pLI 0.48–1.0; between third quantile and 1; and (5) those without a pLI score. For each set, the expected number of LoF heterozygous variants per gene was estimated by the following formula:

$$\text{Expected heterozygous}_{j,k} = L_k \times \frac{\text{mutability}_j}{\sum_{set_k} \text{mutability}_j}$$

where 'j' denotes the 'j$^{th}$' gene, 'k' means the 'k$^{th}$' set, and 'L' means the total number of damaging or LoF heterozygous variants.

## Binomial analysis

Expected and observed counts of rare variants in each gene were compared by independent binomial tests. The expected number of rare de novo damaging variants was determined using the de novo expectation model (see above); the expected number of rare transmitted heterozygous variants was determined using the estimation method mentioned above. Inputs for this test were those with inferred pathogenicity, including D-mis variants (MetaSVM-deleterious or MPC-score ≥ 2) and LoF variants (stop-gains, stop-losses, frameshift insertions, and deletions, or canonical splice-site variants). Binomial analysis for mutational enrichment did not include non-frameshift insertions or deletions, or compound heterozygous variants. The genome-wide significant cut-off was $2.6 \times 10^{-6}$ ( = 0.05/19,347).

## Case-control burden analysis

Case and control cohorts were processed using the same pipeline and filtered with the same criteria. A one-sided Fisher's exact test was used to compare the observed number of total alternative alleles in cases vs. gnomAD controls (without disease-enriched TOPMed samples) for our candidate genes, regardless of transmission pattern.

## Gene lists for specific diseases

Gene lists for VOGM, AVM, CCM, MMD, and human height were curated from the publications referenced and genes were considered disease-causing based on guidelines from the Centers for Mendelian Genomics[23,26,36,113,135–148]. Genes were classified as "possible" VOGM risk genes if they are established VOGM genes, or if they harbored ≥1 damaging DNV in a gene with pLI score ≥0.9 in the gnomAD database.

## Weighted Gene Co-expression Network Analysis (WGCNA)

Robust consensus WGCNA was conducted by applying a processed bulk transcriptome sequencing data set encompassing sixteen human brain regions across human brain development[100]. To increase resolution of the primary neurodevelopmental time periods, analysis was limited to timepoints between gestational week 9 and postnatal year 3. We removed samples >3 standard deviations above the mean network connectivity as outliers. Network analysis was performed with WGCNA[99], assigning genes to specific modules based on bi-weight mid-correlations among genes. Soft threshold power of 10 was chosen to achieve scale-free topology ($r^2 > 0.9$). A signed co-expression network was then generated. The topological overlap matrix was clustered hierarchically using average linkage hierarchical clustering (using '1 − TOM' as a dis-similarity measure). The topological overlap

dendrogram was used to define modules using a minimum module size of 40, a deep split of 4, merge threshold of 0.1.

## Module enrichment analysis

Module gene lists were obtained via WGCNA as described above. In a background set of all genes categorized in co-expression modules, logistic regression was used for an indicator-based enrichment: is.disease ~ is.module + gene covariates (GC content, gene length, and mean expression in bulk RNA-seq atlas)[101]. Of the 88 WGCNA modules, the gray module, by WGCNA convention[100], contains all genes that do not co-express and are consequently unassigned to a co-expression network. Thus, the gray module was excluded from enrichment testing, and enrichment significance was defined at the Bonferroni multiple-testing cutoff ($\alpha = 5.68 \times 10^{-04}$).

## Cell type enrichment analysis

Cell type-enriched genes (cell type markers) were obtained from a scRNA-seq atlas of human brain spanning the period between early fetal development into adulthood and from a scRNA-seq atlas of mouse meninges. In a background set of all genes expressed in ≥ 3 cells of the scRNA-seq atlas, logistic regression was applied for indicator-based enrichment analysis: is.cell.type ~ is.disease + gene covariates (GC content, gene length). All p-values were adjusted by Bonferroni correction ($\alpha = 1.19 \times 10^{-03}$ for the brain parenchyma and $\alpha = 8.33 \times 10^{-03}$ for the meninges).

## Gene Ontology and pathway enrichment analysis

A total of 436 genes, including genes harboring damaging DNVs, LoF-intolerant genes (pLI ≥ 0.9) with rare (Bravo MAF ≤ $5 \times 10^{-5}$) damaging transmitted variants, *EPHB4*, *ACVRL1*, and *ENG* were input into EnrichR R package Version 3.0[149]. Enrichment analysis was performed for gene ontologies (biological processes, cellular components, and molecular functions), and biological pathway (Wiki pathways and Reactome pathways). The top 10 terms with the lowest adjusted p-values were reported.

## Zebrafish husbandry

Adult *Tg(kdr:gfp)$^{zn1}$* zebrafish (AB background) were maintained in 3.0 L tanks with constant water flow (Iwaki Aquatic) under a 14 h/10 h light/dark cycle. They were fed GEMMA Micro 500 (Skretting USA) dry food twice daily supplemented with hatched *Artemia* once daily. Embryos were obtained via natural matings; these embryos were maintained at 28.5° C in E3 medium containing 100 μM 1-phenyl 2-thiourea (PTU) to prevent melanization and facilitate imaging. Embryos were anesthetized with 0.16 mg/mL tricaine (Syndel). Care protocols were developed from techniques developed at Boston Children's Hospital[150,151]. All animal care and handling was performed in accordance with Yale IACUC protocol 2019-20274.

## CRISPR/Cas9 ribonucleoprotein generation, mRNA synthesis, and injection

Gene-specific crRNA sequences were designed using CRISPOR[152] and ordered from IDT. In general, two crRNAs were generated per gene; for paralogous genes such as *ephb4a/b*, both paralogs were targeted simultaneously. These scRNAs were annealed to tracRNA per the manufacturer's instructions (to generate complete sgRNAs), then complexed with Cas9 protein at a 1:2 molar ratio (final concentration of RNA:Cas9 was ~33 μM:66 μM) in Cas9 buffer to generate active ribonucleoprotein knockout reagents. Phenol red was added to a final concentration of 0.1% w/v to generate injection mix. Embryos at the one-cell stage were injected (WPI Pneumatic PicoPump) with 1 nL of injection mix, then allowed to develop to 48 hpf. In some injection mixes these RNPs were combined with rescue or false-rescue mRNAs (200 ng/μL) generated from cloned human cDNAs inserted into pCS2+

and in vitro transcribed using the mMessage mMachine SP6 kit (ThermoFisher) per manufacturer's instructions.

## Validation of knockout reagents

To verify sgRNA efficacy, injected embryos were collected, and crude genomic DNA isolated via boiling/alkaline lysis. Primers flanking the sgRNA target site were used to generate amplicons of ~500 bp. These were denatured, reannealed, digested with T7 endonuclease (NEB) and electrophoresed on 2% agarose. sgRNAs were selected by fragmentation patterns that indicated mismatch DNA arising from CRISPR/Cas9-mediated editing. See Supplementary Table 15 for all crRNAs and primer sets used in this study.

## Microangiography, confocal imaging, and statistical analysis

Anesthetized 48 hpf zebrafish embryos were injected via caudal vein with 1 mg/mL 2,000,000 MW rhodamine dextran (ThermoFisher), embedded in 1.5% agarose, and immediately imaged on a Leica SP8 confocal microscope. Image stacks were processed and analyzed using Fiji/ImageJ. All statistical analysis was performed using R. A Welch two-sample $t$-test was used for all two-level factor analyses; all multi-level factor analyses were by ANOVAs followed by Tukey multiple comparisons of means.

## Plasmids, cell culture, and protein isolation

c-myc-tagged Ephb4 cDNA in pCMV6 was from Origene. Single K650N, R838W and F867L Ephb4 mutations were introduced by site-directed mutagenesis using a QuikChange II XL Site-Directed Mutagenesis Kit (Agilent) per manufacturer's instructions. Cos-7 cells (ATCC) were cultured in DMEM supplemented with 10% FBS and 100 U/ml penicillin/streptomycin (all Thermo Fisher Scientific) in 10 cm culture dishes. At 60% confluency, cells were transfected with 10 μg plasmid using Lipofectamine in Opti-MEM (both Thermo Fisher Scientific). To control for transfection efficiency, cells were co-transfected with 1 μg pEGFP-N1 plasmid (Takara Bio USA). Where indicated, protein stability was assessed by adding 5 mg/ml cycloheximide (Sigma-Aldrich) to the cell growth medium for the indicated times prior to harvest. Cells were harvested 48 h after transfection and washed 2 times with ice-cold PBS. GFP content was assessed by flow cytometry using BD Fortessa or BD FACSCanto instruments (BD Biosciences). Proteins were extracted in NP40 Lysis Buffer (NLB, Thermo Fisher Scientific) by 3 repeats of freeze-thawing on dry ice. Extracts were subsequently centrifuged at 13.000 x $g$. for 10 min at 4 °C and supernatants were used for immunoprecipitation and western blot analysis.

## Immunoprecipitation and Western blotting

For immunoprecipitation, 20 μg total protein in 200 μl NLB was precleared with 10 μl protein A Dynabeads (Thermo Fisher Scientific) for 20 min at 4 °C. Precleared lysates were incubated with 1 μg goat anti-EPHB4 polyclonal IgG (R&D systems, AF446-SP) at 4 °C for 3 h followed by addition of 10 μl Protein A Dynabeads (Invitrogen) for 20 min to capture antibody-protein complexes. Beads were washed 5 times for 5 min each with 200 μl NLB before protein elution by boiling for 10 min in 20 μl 1x NuPage LDS Sample buffer (Thermo Fisher Scientific). For Western blotting, lysates and immunoprecipitates were separated on a Bolt 10% Bis-Tris Plus gel and transferred onto a PVDF membrane (both Thermo Fisher Scientific). To determine total EPHB4 amounts and EPHB4 phosphorylation status, the following antibodies were used: goat anti-EPHB4 polyclonal IgG, mouse anti-phosphotyrosine monoclonal IgG (clone 4G10; MilliporeSigma), rabbit anti-ACTB polyclonal IgG (Cell Signaling Technologies, 4967), rabbit anti-TUBB monoclonal IgG (clone 9F3, Cell Signaling Technologies, 2128), and mouse anti-Myc monoclonal IgG (clone 9E10; EMD MilliporeSigma, OP10-200UG), goat anti-rabbit IgG−HRP (Cell Signaling Technology, 7074), horse anti-mouse IgG−HRP (Cell Signaling Technology, 7076) and donkey anti-goat IgG−HRP (Jackson Immunoresearch, 705-035-147).

## CRISPR generation of the EphB4 F867L mouse

The EphB4 F867L mouse was created via CRISPR/Cas-mediated genome editing[153,154]. Potential Cas9 target guide (protospacer) sequences in exon 15 in the vicinity of the F867 codon were screened using online tool CRISPOR http://crispor.tefor.net, and 5 candidates were selected. Templates for sgRNA synthesis were generated by PCR from a pX330 template, sgRNAs were transcribed in vitro (Megashortscript; ThermoFisher) and purified. sgRNA/Cas9 RNPs were complexed and tested for activity by zygote electroporation, incubation of embryos to blastocyst stage, and genotype scoring of indel creation at the target sites. The gRNA with highest activity was selected to create the knock-in allele. A recombination template oligo (IDT) was designed to create the TTT->CTT codon change and incorporated an EphB4 P865 CCC->CCT silent mutation to destroy the PAM. sgRNA/Cas9 RNP and the template oligo were electroporated into C57Bl/6 J (JAX) zygotes. Embryos were transferred to oviducts of pseudopregnant CD-1 foster females[155]. Genotype screening of tissue biopsies from founder pups was by PCR amplification and Sanger sequencing, followed by breeding to establish germline transmission of the correctly targeted F867L allele. Mice carrying the exon 2-exon 3 floxed alleles of Ephb4 have been described[105]. Cdh5-CreERT2 mice were obtained from Cancer Research UK[156]. All mice were on a mostly C57BL6/J background. Sex of embryos was not determined.

## Mouse genotyping

DNA isolated from tail clips was used for genotyping. The following primers were used to show: (a) the presence of Cre recombinase: forward 5'-ATTTACTGACCGTACACCAAA-3' and reverse 5'-CTGTTTTGCACGTTCACCGGC-3', (b) the Ephb4 floxed allele: forward 5′-GGAATGAGGGCGAGTGGGTT-3′ and the reverse 5′-GGTTGGGGACAAAGAGGAAGA-3, and (c) the Ephb4 F867L mutant locus: forward 5'-GCGAACGATTCTCTCAAGCC-3' and reverse 5'-AGAATAGTGAGGCTGCCGTT-3; for the latter, the obtained product was subsequently digested with EcoN1 restriction enzyme (New England Biolabs) to differentiate wild-type, heterozygous and homozygous mice. 20 μl PCR reactions contained 10 μl GoTaq® green PCR Master mix (Promega), 6 μl nuclease-free water, 0.5 μl of each primer (10 μM), and 3 μl template DNA. Amplification was performed as follows: 98 °C for 2 min, 35 cycles consisting of 98 °C for 30 s, 55 °C for 30 s, and 72 °C for 50 s, followed by 10 min incubation at 72 °C. Products were separated on a 2% agarose gel.

## CD31 whole mount staining of yolk sacs and embryos

EPHB4$^{F867L/+}$ heterozygous mice were intercrossed, and embryos and yolk sacs were harvested at different stages of embryonic development. Embryos and yolk sacs were fixed in 4x diluted Cytofix, Cytoperm (BD Biosciences) and stained with a rat anti-CD31 antibody (BD Biosciences, 550274, 1:50), followed by a secondary donkey anti-rat Alexa Fluor 488 antibody (Invitrogen, 1:300). Specimens were washed in PBS + 0.3% Triton (Sigma-Aldrich). Images were acquired on a Leica SP5 confocal microscope.

## Induced vascular disruption of Ephb4

Ephb4$^{+/F867L}$ heterozygous females were crossed with Ephb4$^{fl/fl}$ Cdh5-CreERT2 males, and pregnant dams were injected i.p. with tamoxifen (Sigma, 0.05 mg/g body weight, dissolved in corn oil) on consecutive days E13.5 and E14.5. Embryos were harvested at E15.5 through E18.5, formalin-fixed, dehydrated and paraffin-embedded. 5 μM sections were stained with H&E or with antibodies against CD31 (SZ31, Dianova, 1:50), LYVE-1 (14917, Abcam, 1:100), active caspase 3 (AF835, R&D Systems, 1:100), or collagen IV (1340-1, Southern Biotech, 1:100), followed by appropriate species-specific fluorochrome-labeled secondary anti-Igs (Jackson Immunoresearch, 1:500). Images were acquired on a Nikon N-SIM + A1R microscope.

**Reporting summary**

Further information on research design is available in the Nature Portfolio Reporting Summary linked to this article.

## Data availability

The raw sequencing data for all VOGM samples in fastq format generated in this study have been deposited in the NCBI database of Genotypes and Phenotypes as well as AnVIL under accession code phs000744.v5.p2 (https://www.ncbi.nlm.nih.gov/projects/gap/cgi-bin/study.cgi?study_id=phs000744.v5.p2). Genomic and phenotypic data for 1,798 unaffected siblings of autism cases and unaffected parents from the SSC can be accessed through SFARI Base (https://base.sfari.org/) with an approved application. The reference genome used in this study is genome assembly GRCh37/hg19 (https://www.ncbi.nlm.nih.gov/datasets/genome/GCF_000001405.13/). Additionally, the data from the mice used in this study can be found in the Supplementary Information/Source Data file. Source data for this paper are provided herein. Source data are provided with this paper.

## Code availability

Software utilized in this study is available at the following web addresses: Samtools v1.3.1 (https://github.com/samtools/samtools); GATK HaplotypeCaller v3.7.0 (https://github.com/broadinstitute/gatk/releases); GATK GenotypeGVCFs v3.7.0 (https://github.com/broadinstitute/gatk/releases); GATK VariantRecalibrator v3.7.0 (https://github.com/broadinstitute/gatk/releases); TrioDeNovo v0.6.0 (http://genome.sph.umich.edu/wiki/Triodenovo); DenovolyzeR v0.2.0 (http://denovolyzer.org); Plink v1.9 (http://pngu.mgh.harvard.edu/-purcell/plink); MetaSVM/CADD13/ANNOVAR v4.2 (http://annovar.openbioinformatics.org); R v3.5.0 (https://www.r-project.org/); Python v2.7 (https://www.python.org/downloads/); EIGENSTRAT v7.2.1 (https://github.com/DReichLab/EIG/tree/master/EIGENSTRAT); EnrichR R package v3.0 (https://cran.r-project.org/web/packages/enrichR/index.html); Monocle R package Version 3 (https://cole-trapnell-lab.github.io/monocle3/); Our in-house pipelines and codes are available at https://github.com/Kahle-Lab/VOGM.

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

## Acknowledgements

This work is supported by the Yale-National Institutes of Health (NIH) Center for Mendelian Genomics (5U54HG006504); R01 NS111029-01A1, R01 NS109358, K12 228168, and the Rudi Schulte Research Institute (K.T.K.); R01 117609 (K.K. and T.J.B.); NIH Medical Scientist Training Program (NIH/National Institute of General Medical Sciences Grant T32GM007205); NIH Clinical and Translational Science Award from the National Center for Advancing Translational Science (TL1 TR001864); the K99/R00 Pathway to Independence Award (K99HL143036 and R00HL143036-02 to S.C.J.), the Vernon W. Lippard Research Fellowship, the Hydrocephalus Foundation Innovator Award (K.T.K. and S.C.J.), the March of Dimes, the Howard Hughes Medical Institute, R01 HL146352 and 2R01 HL120888 (both P.D.K.), and WashU Clinical & Translational Research Funding Program award (CTSA1405 to S.C.J.), and WashU Children's Discovery Institute Faculty Scholar award (CDI-FR-2021-926 to S.C.J.).

## Author contributions

Study design and conceptualization: K.T.K., S.C.J., S.Z, K.Y.M., P.D.K. Cohort ascertainment, recruitment, and phenotypic characterization: K.Y.M., H.S., J.O., J.S., A.J.K., T.D.J., A.B.W.G., P.P., B.J.I., B.A-K., A.T.H., J.M.J., E.J., P.B.S., S-S.L., W.E.B., B.S.C., P.C., C.J.S., A.B.P., G.R., S.S., A.B., A.P.S, T.B., J.Z., K.P.F., J.E.C., M.R.P., E.R.S., M.G., R.P.L., S.C.J., K.T.K. Exome sequencing production and validation: S.M., R.P.L. Exome sequencing analysis: S.Z., X.Z., W.D., D.D., C.G.F., B.C.R., P-Y. F., Y-C.W. Integrative genomics analysis: G.A., S.Z., Q.J.M., E.D., E.K., S.C.J., K.T.K. Phenomics analysis: K.Y.M., M.R., J.S., S.Z. J.O., H.S., A.J.K., T.D.J., S.K., S.M.K. Statistical analysis: S.Z., K.Y.M, G.A., S.C.J., K.T.K. B.L., H.Z., E.Z.E.-O. Sanger sequencing validation: C.N.-W. Neuroimaging characterization: K.T.K., D.B.O., A.M.-D.-L. Structural analysis: J.E.C., T.J.B.

## Competing interests

The authors declare no competing interests.

## Additional information

Shujuan Zhao [1,2,35], Kedous Y. Mekbib [2,3,35], Martijn A. van der Ent [4,35], Garrett Allington [2,5,35], Andrew Prendergast [6], Jocelyn E. Chau [7], Hannah Smith [2,3], John Shohfi [2,3], Jack Ocken [3], Daniel Duran [8], Charuta G. Furey [3,9,10], Le Thi Hao [2], Phan Q. Duy [11], Benjamin C. Reeves [3], Junhui Zhang [12], Carol Nelson-Williams [12], Di Chen [4], Boyang Li [13], Timothy Nottoli [14], Suxia Bai [14], Myron Rolle [2], Xue Zeng [7,15], Weilai Dong [12,15], Po-Ying Fu [1], Yung-Chun Wang [1], Shrikant Mane [12], Paulina Piwowarczyk [16], Katie Pricola Fehnel [16], Alfred Pokmeng See [16], Bermans J. Iskandar [17], Beverly Aagaard-Kienitz [17,18], Quentin J. Moyer [2], Evan Dennis [2], Emre Kiziltug [2], Adam J. Kundishora [3], Tyrone DeSpenza Jr. [3], Ana B. W. Greenberg [2], Seblewengel M. Kidanemariam [19], Andrew T. Hale [20], James M. Johnston [20], Eric M. Jackson [21], Phillip B. Storm [22,23], Shih-Shan Lang [22,23], William E. Butler [2], Bob S. Carter [2], Paul Chapman [2], Christopher J. Stapleton [2], Aman B. Patel [2], Georges Rodesch [24,25], Stanislas Smajda [25], Alejandro Berenstein [26], Tanyeri Barak [3], E. Zeynep Erson-Omay [3], Hongyu Zhao [12,13], Andres Moreno-De-Luca [27], Mark R. Proctor [16], Edward R. Smith [16], Darren B. Orbach [16,28], Seth L. Alper [29], Stefania Nicoli [12,30,31], Titus J. Boggon [7,30], Richard P. Lifton [15], Murat Gunel [3], Philip D. King [4] ✉, Sheng Chih Jin [1,32] ✉ & Kristopher T. Kahle [2,3,33,34] ✉

[1]Department of Genetics, Washington University School of Medicine, St. Louis, MO, USA. [2]Department of Neurosurgery, Massachusetts General Hospital, Harvard Medical School, Boston, MA, USA. [3]Department of Neurosurgery, Yale School of Medicine, New Haven, CT, USA. [4]Department of Microbiology and Immunology, University of Michigan Medical School, Ann Arbor, MI, USA. [5]Department of Pathology, Yale School of Medicine, New Haven, CT, USA. [6]Yale Zebrafish Research Core, Yale School of Medicine, New Haven, CT, USA. [7]Department of Molecular Biophysics and Biochemistry, Yale School of Medicine, New Haven, CT, USA. [8]Department of Neurosurgery, University of Mississippi Medical Center, Jackson, MS, USA. [9]Department of Neurosurgery, Barrow Neurological Institute, Phoenix, AZ, USA. [10]Ivy Brain Tumor Center, Department of Translational Neuroscience, Barrow Neurological Institute, Phoenix, AZ, USA. [11]Department of Neurosurgery, University of Virginia School of Medicine, Charlottesville, VA, USA. [12]Department of Genetics, Yale School of Medicine, New Haven, CT, USA. [13]Department of Biostatistics, Yale School of Public Health, New Haven, CT, USA. [14]Yale Genome Editing Center, Department of Comparative Medicine, Yale School of Medicine, New Haven, CT, USA. [15]Laboratory of Human Genetics and Genomics, The Rockefeller University, New York, NY, USA. [16]Department of Neurosurgery, Boston Children's Hospital, Harvard Medical School, Boston, MA, USA. [17]Department of Neurological Surgery, University of Wisconsin School of Medicine and Public Health, Madison, WI, USA. [18]Department of Radiology, University of Wisconsin School of Medicine and

Public Health, Madison, WI, USA. [19]Department of Biochemistry, Microbiology and Immunology, University of Ottawa, Ottawa, ON, Canada. [20]Department of Neurosurgery, University of Alabama School of Medicine, Birmingham, AL, USA. [21]Department of Neurosurgery, Johns Hopkins University School of Medicine, Baltimore, MD, USA. [22]Department of Neurosurgery, Hospital of the University of Pennsylvania, Philadelphia, PA, USA. [23]Division of Neurosurgery, Children's Hospital of Philadelphia, Philadelphia, PA, USA. [24]Service de Neuroradiologie Diagnostique et Thérapeutique, Hôpital Foch, Suresnes, France. [25]Department of Interventional Neuroradiology, Hôpital Fondation A. de Rothschild, Paris, France. [26]Department of Neurosurgery, Icahn School of Medicine at Mount Sinai, New York, NY, USA. [27]Department of Radiology, Autism & Developmental Medicine Institute, Genomic Medicine Institute, Geisinger, Danville, PA, USA. [28]Department of Neurointerventional Radiology, Boston Children's Hospital, Harvard Medical School, Boston, MA, USA. [29]Division of Nephrology and Center for Vascular Biology Research, Beth Israel Deaconess Medical Center, and Department of Medicine, Harvard Medical School, Boston, MA, USA. [30]Department of Pharmacology, Yale School of Medicine, New Haven, CT, USA. [31]Yale Cardiovascular Research Center, Department of Internal Medicine, Section of Cardiology, Yale School of Medicine, New Haven, CT, USA. [32]Department of Pediatrics, Washington University School of Medicine, St. Louis, MO, USA. [33]Division of Genetics and Genomics, Boston Children's Hospital, Boston, MA, US. [34]Broad Institute of MIT and Harvard, Cambridge, MA, USA. [35]These authors contributed equally: Shujuan Zhao, Kedous Y. Mekbib, Martijn A. van der Ent, Garrett Allington. ✉e-mail: kingp@umich.edu; jin810@wustl.edu; Kahle.Kristopher@mgh.harvard.edu

