## [Peer Review File · Nature Communications]

Mutation of key signaling regulators of cerebrovascular development in vein of Galen malformationsREVIEWER COMMENTS

Reviewer #1 (Remarks to the Author):

In this manuscript by Zhao et al., the authors identify novel genetic variants and confirm the major role of RASA1 and EphB4 mutations in vein of Galen malformation.

The authors use exome sequencing and computational genetics as well as integrative genomics analysis to validate novel and previous data regarding VOGM. They highlight a potential convergence of the different VOGM genes (previous and newly discovered ones in this study) towards the Ras-MAPK signaling pathway. They propose a specific enrichment of VOGM genes in endothelial cells. Finally, they validate some of the novel genes and EphB4 variants in VOGM using zebrafish and mice models.

This work is interesting as it unravels for the first time novel mutations responsible for VOGM, highlights the role of these and previous identified mutations in EC development and identifies Ras/ERK/MAPK signaling as a common pathway linking these genes.

Even though some of the findings here have already been published and discussed previously, however, the holistic approach that the authors use, the identification of novel mutations and their validation in vivo brings new perspectives for VOGM understanding and potential treatment.

While the genetic study is overall solid, well presented (see minor issues) and quite convincing, some of the in vivo and biochemical data might require a more thorough analysis and better quantification to make this work complete.

Upon revision, this paper will make a good contribution to the field of VOGM.

Major issues

Biochemical analysis

Western blot analysis could be improved since most of it is overexposed and quantifications are missing. This is true for panels E-G in Figure 1 as well as panel D in Figure 4.

As for Figure 1, we would recommend using a different control protein, instead of ActB, that would resist longer to CHX treatment. This would allow a better and more accurate analysis of EphB4 decay. We are slightly puzzled by the i.p. analysis in panel G, for instance the phosphorylation rate seems to be pretty high in comparison to the total EphB4 that is quite weak. Using anti-myc antibody instead might solve this issue.

As for Figure 4, apart from the fact that the western blot is totally overexposed, some controls (e.g. EphB4+/+) are missing as well as the quantification of phosphorylation rate. Authors use liver lysates to assess MAPK activation, but due to the heterogeneity of this tissue, is it possible to assess and quantify this in a different tissue to consolidate the conclusion?.

In vivo analysis

Authors validate some of the mutations they characterize using zebrafish. Even though Figure 2 panel D is quite helpful, however, the images in panels E,F and G are small, highly pixelized and the reader cannot identify the underlying defects. The dextran injection/labeling is unnecessary and does not help as this is too bright, arrows and dashed lines are not clear. The quantifications in supplementary figure 7 are really nice and add important information. It would be better if authors would split this image in 2, 1/ ephb4 results including related supp. information and 2/ acvrl1 and itgb1 results and related supp. information. Note that Supp. Figure 7 include itgb1 in the text but not in the graphs in C.

F0 Crispr injections are highly mosaic for indels, it is therefore important to organize phenotypes into Normal, Slight, Medium and Severe that would represent better the variability of F0 analysis. In line 450, the authors mention that "histological examination revealed extensive apoptotic cell death (Fig. 4A)". However, this is not shown in Figure 4 (or anywhere else). Authors should either examine this (e.g. Caspase 3 labeling), add it to the results or disregard this information.

In Panel C Figure 4, Images from Ephb4 +/+ are missing. This is quite important since the heterozygous embryo clearly seems to have abnormal vascularization of the yolk sac (Panels A and B) and shown embryo is smaller than control (a scale bar is missing). Indeed, the authors mention in the text (lines 440-41), "Heterozygous mice were viable, fertile and exhibited no obvious vascular developmental or other defects". This statement is not totally accurate, a more thorough analysis is recommended and this issue might need to be quantified.

Figure 4 Panel B, it is not obvious that the bottom right image is a higher magnification of the bottom left one (scale bars are missing too). It would be useful to the reader if a similar higher magnification of a control yolk sac is added so to directly compare vascularization.

Figure 1 Panel A could be improved, there is no indication of specimen orientation, a scale bar is missing, asterisks and arrows are too small. The genders and ages of represented control and

patient are not mentioned.

Minor issues

In Figure 3, it would be useful if authors would highlight 'Regulation of angiogenesis' in panel C since this is directly relevant to this work. Even though this might not be directly relevant but it is interesting to mention that AVM and AVM-VOGM sets in panel D are enriched in microglia. Any possible interpretation?

In Discussion (line 535), the sentence "Our data in mice supports this genetic mechanism, as heterozygous Ephb4F867L mice expressing a VOGM-specific EPHB4 kinase-domain missense variant exhibited severe vascular abnormalities, but only when carrying a second variant allele" is not clear. It would be useful if authors could mention which allele is that.

In Supplementary Table 2, only 1/3 of the cohort is of Female gender. Is that a bias? Does it mean that VOGM is more frequent in men, more deleterious? It would be useful to clarify this in the text. Supplementary Figure 10 is of poor quality, one cannot read it.

In Material and Methods, line 656, please define DNV abbreviation.

In Supplementary Figure 3, it would be interesting to point out the described NOTCH1 mutation in a cartoon similar to other variants.

In Supplementary Table 4, it would be useful to define D and T abbreviations.

Supplementary Figure 8 might not be necessary.

Reviewer #2 (Remarks to the Author):

Zhao et al identified novel de novo and transmitted variants in patients with vein of Galen malformations (VOGM) by performing whole exome sequencing of a large number of proband-trio cohorts. They identified loss-of-function de novo variants in RASA1, as well as damaging transmitted variants in EPHB4. They also found pathogenic variants in ACVRL1, NOTCH1, ITGB1 and PTPN11. Integrated analyses revealed enriched expression of VOGM genes in developing endothelial cells using available single-cell RNA-seq of human cerebrovascular cells. The authors generated zebrafish models of some loss-of-function VOGM genes, including Ephb4, Acvrl1, and Itgb1, and observed abnormal vascular phenotypes. Finally, the authors generated an inducible mouse model expressing EPHB4 F867L in endothelial cells and observed vascular defects. Overall, the study suggested that damaging de novo variants account for > 12% of VOGM cases and highlighted the importance of the RAS/ERK/MAPK signaling axis in VOGM pathogenesis. The findings of this study are novel and reveal a new perspective on genes that are likely involved in this disease. While the data for the most part support the conclusions, there are several questions that remain.

Major Comments:

1) While the clinical studies were comprehensive, the animal models require additional work. In the zebrafish studies, it is not clear why the authors did not try to rescue the ephb4a/b phenotype as they did with the other models. Did intracerebral hemorrhage occur in the zebrafish models? The itgb1a/b phenotype is not very convincing. Did the authors use blinding to score phenotypes? For the wild-type animals, were they injected with all of the same CRISPR components, but with a non-targeting guide RNA? The cerebrovasculature is very sensitive to injection and this control is essential for this type of study.

2) The weakest part of the paper is the connection to elevated MAPK/ERK signaling. It would have been helpful to show that cells transfected with the EPHB4 mutants have defective MAPK/ERK signaling in Figure 1. It was surprising that the pTyr blot showed reduced phosphorylation in the total lysate. An uncropped blot would be helpful to show that there is increased phosphorylation of other proteins (unless this protein does not act as a dominant negative). Showing pERK levels in cells over-expressing the identified mutations in ACVRL1 and ITGB1 in vitro would further support the model. Additional data is needed to support the conclusion that MAPK/ERK signaling is elevated in the endothelium of EPHB4 mutant mice. Currently, there is only a western blot on total liver lysate. The authors should show pERK staining in the vasculature, particularly in the cerebrovasculature.

3) The EPHB4 mutant mouse models are not fully characterized. In the heterozygous animals, the phenotypes should be quantified. It appears that the animals are smaller and have less complex vascular networks in the yolk sac vasculature. Wild-type embryos should be shown in Figure 4C.

For the mice with global mutation plus endothelial-specific mutation, are there any phenotypes that resemble VOGM? For example, are there any arteriovenous malformations or vessel dilations? What does the cerebrovasculature look like? This model is interesting, but the data presented appears to be preliminary. Phenotypes should be assessed after activating Cre at different stages to better understand when EPHB4 might be contributing to VOGM-like phenotypes. The hemorrhage phenotype is not that informative.

Minor Comments:

- 1) What is the mutant *Itgb1* $\Delta 10$ mRNA that was injected in *Itgb1a/b* depleted zebrafish? Is it similar to the two damaging variants identified in two separate probands?
- 2) The title of Table 1 indicates 114 VOGM cases, but in the table itself, there are 90 cases listed. This should be clarified.
- 3) In Supplementary Figure 7C graphs, the x-axis mutant label is incorrect. This should be *itgb1a/b* mutants and not *acvrl1*. Also, in the figure text for panel B and C, 'eph4a/b-depleted' should be replaced by the correct depleted zebrafish model.
- 4) In the 3rd paragraph of the main text under "VOGM risk genes converge in growth factor-regulated Ras/ERK/MAPK signaling networks in developing cerebral endothelial cells" section, it should be referencing Fig. 3D instead of Fig. 3E.
- 5) In the 1st and 2nd paragraph of the main text under "Variants in ACVRL1 and other mutation-intolerant Mendelian vascular disease genes", it should be "R-hsa" instead of "R-has".
- 6) In Supplementary Figure 3A, it would be helpful to show the pedigree for clarity.
- 7) Can the authors find a different way to refer to the scRNA-seq modules? Referring to color isn't really helpful. Is there a diagram of the relationship of the modules that includes these colors?
- 8) Was the normal and arteriovenous malformation scRNA-seq data used from the Winkler et al. paper or just the normal dataset? This was not clear. Was there a difference in enrichment for VOGM disease genes in the normal versus malformed vasculature?

Reviewer #3 (Remarks to the Author):

Zhao et al identify new risk genes for vein of Galen malformations (VOGMs), which are severe congenital brain arteriovenous malformations. Using trio-based WES data of a large VOGM cohort of 310 proband-family exomes, they identify pathogenic variants in *RASA1*, *EPHB4*, *ACVRL1*, *PTNPN11*, and *NOTCH1*. Integrated bioinformatic analysis of human cerebrovasculature single-cell transcriptomes to define co-expression modules, cell types, and biological pathways show enrichment of VOGM genes in endothelial cells, and reveals enrichment of pathways related to growth factor-regulated Ras/MAPK/ERK signaling in VOGM. Finally, some of the candidate genes were validated in vivo by characterizing vascular phenotypes in zebrafish or mouse carrying a VOGM gene mutation.

The first part of the study is based on an impressive patient cohort and provides highly valuable insight into genetic heterogeneity of and risk genes for VOGM. However, the in vivo validation of VOGM genes is too superficial to provide mechanistic understanding of disease pathogenesis, or to extend our current understanding of the role of these genes in regulating normal vascular development. To my mind the data would be stronger if focused on the genetic findings. This would also allow improving the clarity of the presentation of the clinical data. Alternatively, the authors should provide more in-depth analysis of disease pathogenesis by focusing on selected genetic mutation(s), taking into account that all of the identified VOGM risk genes have already been extensively studied in endothelial cells and in the context of vascular development and disease.

Major comments:

1. Can the authors provide direct evidence for abnormal activity of Ras/MAPK/ERK in endothelial cells from VOGMs, to justify the statement in the title? For example, by staining of tissue biopsies, or WB analysis of patient-derived ECs?
2. Methods such as Module and GO analysis, Reactome pathway analysis, and STRING analysis can

provide valuable insight and a basis for generating hypotheses, but the results should be interpreted with caution and confirmed using other methods. Here the authors present several figures with such data, without additional validation of the conclusions drawn from these data.

3. The value of using two powerful single-cell RNA seq datasets in the study is unclear. The main conclusion is that ECs express genes associated with VOGM, but this is already known from previous studies. All of the identified VOGM risk genes have been extensively studied in endothelial cells and in the context of vascular development and disease. Additional data obtained from the module analysis is interesting but suggestive only. Please explain the color coding used for the module analysis.

4. Figure 2, Zebrafish phenotypes: do all images represent the same depth/part of the brain, for example in Figure 2E? I note missing flow in the smaller vessels in the central arteries. Is it possible that vein enlargement is secondary to impaired/disturbed flow? Better characterization of the phenotypes, including a developmental time course, would be needed to understand the mechanism of pathogenesis.

5. Page 11: "...the possible requirement of EPHB4 kinase activity for normal vascular development is unknown". Kinase inactivating mutations in EPHB4 were previously described in non-immune hydrops fetalis and lymphedema (PMID: 27400125). The authors should discuss these findings in the context of theirs.

6. Figure 4, VOGM-specific EPHB4 mutant mouse: The authors focus on cutaneous lesions, which they mention were not observed in the VOGM patients. Do these EPHB4 mutants also recapitulate lesions in the brain? Figure 4E: Is this hemorrhage, or blood-filled lymphatic vessels? Increased pERK should be shown in ECs (as I understand now shown in total liver lysate). Please show pAKT as well (also enriched in VOGM based in module analysis in Figure 3).

Other comments:

1. Graphical abstract has too much detailed information and does not summarize key findings of the study.

2. Figure panels should be labelled in the order they are referred to in the text (for example Figure 2E vs 2F)

3. Page 14: "...define developing endothelial cells as a critical spatio-temporal locus of VOGM pathophysiology. This appears to be different from other cerebral vascular anomalies such as Moyamoya disease and cerebral cavernous malformations". To my knowledge, CCM gene mutations in ECs drive cerebral cavernous malformations. Please comment.

4. Supplementary Figure 1: Duplicated image KVOGM 65-1 and KVOGM 70-1.

5. Supplementary Figure 7C: Figure legends states ITGB1, figure acvr11

6. Cdh5-ert2cre – should it be Cdh5-creERT2?

7. Especially in the results, it was not always clear which data are novel, and what has been shown previously. One example is below, but authors are recommended to check the entire manuscript carefully and refer to previous findings where applicable:

Page 11: "Based on our human genetic (Fig. 1) and integrative genomic results (Fig. 3), we hypothesized that genetic inactivation of a kinase-dependent, EPHB4-regulated RASA1 signaling mechanism in endothelial cells causes constitutive Ras-MAPK activation and disruption of VEGF-associated developmental angiogenesis in vivo." As the authors discuss later, EPHB4/RASA1 deficiency has been shown to result in increased Ras/MAPK activation and the relevant studies should be referred to in this context.

REVIEWER COMMENTS

Reviewer #1 (Remarks to the Author):

In this manuscript by Zhao et al., the authors identify novel genetic variants and confirm the major role of RASA1 and EphB4 mutations in vein of Galen malformation.

The authors use exome sequencing and computational genetics as well as integrative genomics analysis to validate novel and previous data regarding VOGM. They highlight a potential convergence of the different VOGM genes (previous and newly discovered ones in this study) towards the Ras-MAPK signaling pathway. They propose a specific enrichment of VOGM genes in endothelial cells. Finally, they validate some of the novel genes and EphB4 variants in VOGM using zebrafish and mice models.

This work is interesting as it unravels for the first-time novel mutations responsible for VOGM, highlights the role of these and previous identified mutations in EC development and identifies Ras/ERK/MAPK signaling as a common pathway linking these genes.

Even though some of the findings here have already been published and discussed previously, however, the holistic approach that the authors use, the identification of novel mutations and their validation in vivo brings new perspectives for VOGM understanding and potential treatment.

While the genetic study is overall solid, well presented (see minor issues) and quite convincing, some of the in vivo and biochemical data might require a more thorough analysis and better quantification to make this work complete.

Upon revision, this paper will make a good contribution to the field of VOGM.

Response: We appreciate the reviewer's positive feedback and valuable critique. We have addressed each comment point-by-point, below.

Major issues

Biochemical analysis

Western blot analysis could be improved since most of it is overexposed and quantifications are missing. This is true for panels E-G in Figure 1 as well as panel D in Figure 4.

Response: We have improved our western blot analysis, see below:

Fig. 1E. Weaker exposures of the previous c-myc and tubulin blots are provided, and signals are quantified. An additional experiment has been performed to show blotting for EPHB4, and the results are quantified. To save space, flow cytometry data showing equivalent transfection efficiency is summarized at the top of each sub-panel. The original flow cytometry plots for each experiment are provided in Supplemental Figure 3. In summary, the additional data provided confirm that EPHB4 kinase mutations do not consistently affect protein stability in the steady state.

Fig. 1F. Weaker exposures of blots are provided and quantitated. In the previous version of the manuscript, we inadvertently provided images of SLC25A9 decay instead of actin B decay for the WT and F867L transfection experiments. The correct images are now provided. In summary, there is no obvious difference in the rate of decay of EPHB4 kinase mutants compared to WT EPHB4.

Fig.1G. A weaker exposure of the pTyr blot is provided and results are quantitated.

Fig. 4D. has been dropped from the paper.

As for Figure 1, we would recommend using a different control protein, instead of ActB, that would resist longer to CHX treatment. This would allow a better and more accurate analysis of EphB4 decay.

Response: As now shown, Act B is stable throughout the time course in transfection experiments (see comment above).

We are slightly puzzled by the i.p. analysis in panel G, for instance the phosphorylation rate seems to be pretty high in comparison to the total EphB4 that is quite weak. Using anti-myc antibody instead might solve this issue.

Response: We have provided a weaker exposure of the pTyr blot. As expected, the strength of the signal is a function of exposure time. A comparison of signal intensity between different blots that use different antibodies and exposure times would not be valid.

As for Figure 4, apart from the fact that the western blot is totally overexposed, some controls (e.g. EphB4+/+) are missing as well as the quantification of phosphorylation rate. Authors use liver lysates to assess MAPK activation, but due to the heterogeneity of this tissue, is it possible to assess and quantify this in a different tissue to consolidate the conclusion?

Response: Fig. 4D. has been dropped from the paper. Further analysis has shown that the differences between embryos of different genotypes are not consistent. While we retain the view that phenotypes in Ephb4 fl/F867L Cdh5 ert2cre embryos are ultimately a consequence of dysregulated Ras-MAPK signaling, this has been difficult to demonstrate convincingly by Western blotting or immunofluorescence staining of tissue sections. Consequently, we have moderated these claims in the resubmitted manuscript. We have nonetheless provided additional data that the EPHB4 F867L mutation phenocopies EPHB4 and RASA1 loss mutations in mice. New Figs. 5 and 6 provide more detailed information on the evolution of vascular phenotypes following the switch to exclusive expression of EPHB4 F867L in the vasculature at E13.5. In summary, the first event is the back-filling of lymphatics with venous blood that likely reflects breakdown of lympho-venous valves. This is first observed in some embryos at E16.5 (images at E17.5 are shown in Fig. 5). At E17.5, apoptotic blood vascular endothelial cells are readily identified in the skin and brain, with clear signs of hemorrhage in these organs. Collagen IV also accumulates within vascular endothelial cells of the skin. By E18.5, severity of vascular hemorrhage has increased in skin, and lymphatics are absent. We have reported these phenotypes previously in EPHB4-null and RASA1-null embryos.

In vivo analysis

Authors validate some of the mutations they characterize using zebrafish. Even though Figure 2 panel D is quite helpful, however, the images in panels E,F and G are small, highly pixelized and the reader cannot identify the underlying defects. The dextran injection/labeling is unnecessary and does not help as this is too bright, arrows and dashed lines are not clear.

Response: We agree that the original panels were compressed. We have reorganized the figure to better emphasize the angiogram results. We believe the dextran labeling remains informative. Firstly, angiogram labeling helps identify blood vessels without transgenes, which can confound data interpretation through ectopic expression. Secondly, angiograms identify blood-filled vessels as opposed to nascent endothelial cell contacts which may or may not develop into functional vessels. As we use this criterion in our quantification, we consider it important to retain this information in the figure.

The quantifications in supplementary figure 7 are really nice and add important information. It would be better if authors would split this image in 2, 1/ ephb4 results including related supp. information and 2/ acvrl1 and itgb1 results and related supp. information.

Response: We agree that the quantification is important. We have split this image across two figures (Supp. Figure 8 and Supp. Figure 9).

Note that Supp. Figure 7 include itgb1 in the text but not in the graphs in C.

Response: We have removed *Itgb1* data in the Figure (see below).

F0 Crispr injections are highly mosaic for indels, it is therefore important to organize phenotypes into Normal, Slight, Medium and Severe that would represent better the variability of F0 analysis.

Response: We agree that F0 analyses are inherently mosaic. We would, however, emphasize that F0 analysis is increasingly recognized as useful due to the very high efficiency of CRISPR/Cas9 ribonucleoprotein complexes. Our use of multiple guides in these experiments leads to DNA editing probabilities approaching 1 in each cell (PMID:33416493). We also appreciate the reviewer's suggestion to organize phenotypes categorically. However, we consider our quantification of these experimental data (continuous physical parameters such as PCS are, etc.) more rigorous and useful than categorization. We note that any categorization, including determination of the number of categories, may be in part subjective, whereas we report quantitative aspects measurable in F0.

In line 450, the authors mention that "histological examination revealed extensive apoptotic cell death (Fig. 4A)". However, this is not shown in Figure 4 (or anywhere else). Authors should either examine this (e.g. Caspase 3 labeling), add it to the results or disregard this information.

Response: The statement has been dropped from the paper.

In Panel C Figure 4, Images from Ephb4 +/+ are missing. This is quite important since the heterozygous embryo clearly seems to have abnormal vascularization of the yolk sac (Panels A and B) and shown embryo is smaller than control (a scale bar is missing).

Response: Images from an E9.5 EphB4+/+ embryo are now provided for comparison (now Fig. 4E). We have provided magnified images of Ephb4 +/+ and Ephb4+/F867L heterozygous yolk sacs in the new Fig. 4D. Vascularization of the heterozygous yolk sac is normal. Scale bars have been added to the embryo images in Fig. 4B. In addition, we have provided a graphic showing the mean size of multiple E10.5 embryos of the different genotypes in a new Fig. 4C. There is no difference in size between heterozygous and wild-type embryos.

Indeed, the authors mention in the text (lines 440-41), "Heterozygous mice were viable, fertile and exhibited no obvious vascular developmental or other defects". This statement is not totally accurate, a more thorough analysis is recommended, and this issue might need to be quantified.

Response: We have included a new Figure 4A that displays the number of live-born mice resulting from crosses of heterozygotes. The number of live-born heterozygotes is approximately twice the number of live-born wild-type mice, consistent with the expected outcome for a mutation which in heterozygous form lacks detrimental effects. Moreover, these heterozygotes have exhibited grossly normal phenotype throughout their lifespan.

Figure 4 Panel B, it is not obvious that the bottom right image is a higher magnification of the bottom left one (scale bars are missing too). It would be useful to the reader if a similar higher magnification of a control yolk sac is added so to directly compare vascularization.

Response: Scale bars and higher magnifications of wild type and het yolk sacs have been added to Fig. 4E.

Figure 1 Panel A could be improved, there is no indication of specimen orientation, a scale bar is missing, asterisks and arrows are too small. The genders and ages of represented control and patient are not mentioned.

Response: We have modified Figure 1 Panel A with larger annotations, a scale bar and new orientation marks. The figure legend also now includes age and sex information of the children.

Minor issues

In Figure 3, it would be useful if authors would highlight 'Regulation of angiogenesis' in panel C since this is directly relevant to this work.

Response: "Regulation of angiogenesis" in panel C has been highlighted.

Even though this might not be directly relevant, but it is interesting to mention that AVM and AVM-VOGM sets in panel D are enriched in microglia. Any possible interpretation?

Response: We note that the AVM gene set in panel D is enriched with microglia, but the AVM-VOGM gene set is also enriched. The major contributing signal comes from the AVM gene set, as the VOGM gene set alone did not show significant enrichment. Evidence for inflammatory cell involvement in AVM has been previously shown, where AVM tissues displayed more neutrophil and macrophage/microglia markers than epilepsy control tissues (PMID:18825001). Our data show that the AVM gene set is enriched in microglia, supporting involvement of inflammation in the clinical course of the disease state; however, it remains unclear whether inflammation is involved in the pathogenesis of the lesion.

In Discussion (line 535), the sentence "Our data in mice supports this genetic mechanism, as heterozygous Ephb4F867L mice expressing a VOGM-specific EPHB4 kinase-domain missense variant exhibited severe vascular abnormalities, but only when carrying a second variant allele" is not clear. It would be useful if authors could mention which allele is that.

Response: We have further clarified this mouse experiment at the appropriate point.

In Supplementary Table 2, only 1/3 of the cohort is of Female gender. Is that a bias? Does it mean that VOGM is more frequent in men, more deleterious? It would be useful to clarify this in the text.

Response: We note that this study is a non-randomized, gene-discovery study utilizing all available samples, rather than a randomized trial. As a result, the sex ratio in our study does not necessarily reflect the true sex ratio in the actual VOGM population. We have further clarified this in the text.

Supplementary Figure 10 is of poor quality, one cannot read it.

Response: Figure 10 has been reorganized.

In Material and Methods, line 656, please define DNV abbreviation.

Response: DNV abbreviation in line 656 has been defined.

In Supplementary Figure 3, it would be interesting to point out the described NOTCH1 mutation in a cartoon similar to other variants.

Response: A cartoon for the NOTCH1 mutation now shown in Supp. Figure 4. We also added protein labels to make it clearer.

In Supplementary Table 4, it would be useful to define D and T abbreviations.

Response: D and T abbreviations have been defined in Supplementary Table 4.

Supplementary Figure 8 might not be necessary.

Response: We agree with the reviewer that Supplementary Figure 8 might not be necessary. This figure has been removed.

Reviewer #2 (Remarks to the Author):

Zhao et al identified novel de novo and transmitted variants in patients with vein of Galen malformations (VOGM) by performing whole exome sequencing of a large number of proband-trio cohorts. They identified loss-of-function de novo variants in RASA1, as well as damaging transmitted variants in EPHB4. They also found pathogenic variants in ACVRL1, NOTCH1, ITGB1 and PTPN11. Integrated analyses revealed enriched expression of VOGM genes in developing endothelial cells using available single-cell RNA-seq of human cerebrovascular cells. The authors generated zebrafish models of some loss-of-function VOGM genes, including Ephb4, Acvrl1, and Itgb1, and observed abnormal vascular phenotypes. Finally, the authors generated an inducible mouse model expressing EPHB4 F867L in endothelial cells and observed vascular defects. Overall, the study suggested that damaging de novo variants account for > 12% of VOGM cases and highlighted the importance of the RAS/ERK/MAPK signaling axis in VOGM pathogenesis. The findings of this study are novel and reveal a new perspective on genes that are likely involved in this disease. While the data for the most part support the conclusions, there are several questions that remain.

Response: We appreciate the reviewer's positive feedback and valuable critique. We have addressed each comment below in a point-by-point manner.

Major Comments:

1) While the clinical studies were comprehensive, the animal models require additional work. In the zebrafish studies, it is not clear why the authors did not try to rescue the ephb4a/b phenotype as they did with the other models.

Response: We chose not to repeat these experiments due to the prior work on ephB4 in zebrafish performed by Vivanti et al. (PMID: 29444212); we used this gene primarily as a positive control for phenotypes.

Did intracerebral hemorrhage occur in the zebrafish models?

Response: Though intracerebral hemorrhage (ICH) was not formally scored, we judged that ICH was not increased in our zebrafish models, based on our extensive prior experience (PMID: 34887573).

The itgb1a/b phenotype is not very convincing. Did the authors use blinding to score phenotypes?

Response: We took several measures to ensure unbiased analysis of our data. Firstly, we blinded our existing dataset and assigned data analysis to an independent scientist. Despite maintaining integrity of the observed trends, the analysis revealed an effect size below the significance threshold. In combination with the data obtained from acvrl1 scramble experiments (described below), we concluded that assessing A/V contacts as a metric would be a substantial challenge. Previous research has established the essential role of β 1 integrin in forming stable, non-leaky blood vessels during vessel maturation, in promoting endothelial sprouting and in localizing VE-cadherin at endothelial junctions (PMID: 25752958). Constitutive targeting of Itgb1 in endothelial cells was incompatible with normal morphogenesis of the embryonic vasculature patterning and postnatal vascular remodeling (PMID: 17984225). Loss of Itgb1 in nascent endothelium also disrupted arterial endothelial cell polarity and lumen formation (PMID: 20152176).

To investigate the role of *Itgb1* in lumen formation, we used IMARIS to measure vascular volume in easily identifiable vessels (BA, PHBC, LDA, DA, PCS) in zebrafish injected with *itgb1a/b* or with scrambled gRNAs, and in uninjected fish. Volume was significantly affected solely in the *itgb* condition in PHBC, LDA, and DA (see accessory figure). The automated nature of IMARIS-mediated volume detection decreases the likelihood that blinding would substantially affect these results.

We conclude from these results that the phenotype of *itgb1a/b* knockout is relatively mild. Consequently, we have removed these results from the main figures, and do not plan to include the *itgb1* zebrafish data in the manuscript. However, we have included these results here for transparency and completeness.

For the wild-type animals, were they injected with all of the same CRISPR components, but with a non-targeting guide RNA? The cerebrovasculature is very sensitive to injection and this control is essential for this type of study.

Response: We agree that this is an important control. We have added data from scrambled *acvr1* gRNA injections as negative control. Generally, we find that this does affect A/V contacts but does not affect the increased PCS/PHBC volume phenotypes we observe in the *acvr1* injections. We have updated the figures accordingly to emphasize this control experiment. We feel this data strengthens the validity of the *acvr1* phenotype (Figure 2).

2) The weakest part of the paper is the connection to elevated MAPK/ERK signaling. It would have been helpful to show that cells transfected with the EPHB4 mutants have defective MAPK/ERK signaling in Figure 1.

Response: The purpose of Fig. 1 is to show that the EPHB4 mutants are stable and kinase-inactive. As far as MAPK activation is concerned, EPHB4 can act as an inhibitor of Ras-MAPK signaling triggered by other growth factor receptors, but only in non-transformed endothelial cells (PMID: 9990854; 10518221). Therefore, the reviewer's suggested experiment would require primary endothelial cells such as HUVEC. However, it would not suffice to express mutants in HUVEC, as HUVEC express endogenous wild-type EPHB4. Thus, stimulation with Ephrin B2 ligand would engage both endogenous wild-type and transfected EPHB4, and Ras-MAPK inhibition would still result regardless of the type of EPHB4 transfected. In this regard, please also consider that, as evidenced by the viability of the heterozygous EPHB4 F867L mice, kinase-dead EPHB4 mutants do not act in a dominant-negative fashion. One could attempt to ligand-stimulate only transfected, epitope-tagged EPHB4 in HUVEC; however, EPHB4 receptor multimerization would lead to co-liganding of the endogenous wild-type EPHB4 with this approach. The only feasible approach to the suggested experiment would be to knock out endogenous EPHB4 in HUVEC and then re-express wild-type or mutant EPHB4. This would necessitate biallelic CRISPR targeting and homology-directed repair. While such an event might occur in a small percentage of cells, their selection and expansion would pose a challenge. We hope the reviewer agrees that this type of experiment is technically demanding and beyond the scope of the current manuscript.

It was surprising that the pTyr blot showed reduced phosphorylation in the total lysate. An uncropped blot would be helpful to show that there is increased phosphorylation of other proteins (unless this protein does not act as a dominant negative).

Response: The mutants show essentially no pTyr phosphorylation, as expected of kinase-dead mutants, since pTyr phosphorylation of EPHB4 is mediated by EPHB4 itself (auto-phosphorylation).

Showing pERK levels in cells over-expressing the identified mutations in ACVRL1 and ITGB1 in vitro would further support the model. Additional data is needed to support the conclusion that MAPK/ERK signaling is elevated in the endothelium of EPHB4 mutant mice. Currently, there is only a western blot on total liver lysate. The authors should show pERK staining in the vasculature, particularly in the cerebrovasculature.

Response: We encountered challenges in demonstrating increased MAPK activation in the vasculature of EPHB4 F867L embryos by immunostaining of tissue sections. While we still believe that this activation is the underlying cause of vascular defects, we acknowledge that we cannot definitively prove it at this stage. We have therefore moderated this claim in the revised manuscript. However, our further characterization of the model shows that it phenocopies EPHB4-null and RASA1-null mutant models (new Figs. 4-6).

3) The EPHB4 mutant mouse models are not fully characterized. In the heterozygous animals, the phenotypes should be quantified. It appears that the animals are smaller and have less complex vascular networks in the yolk sac vasculature.

Response: We have provided further characterization in Figs. 4-6.

New Fig 4A presents numbers of live-born mice from crosses of hets. The number of live-born hets is approximately twice the number of live-born wild-type mice, expected for a mutation that in heterozygous form exhibits no deleterious effects. Hets remain healthy throughout life, and we have not observed any phenotype in these mice of any type. In addition, we have provided a graphic showing the mean size of multiple E10.5 embryos of the different genotypes in a new Fig. 4C. Heterozygous and wild-type embryos do not differ in size. Higher magnifications of wild-type and het yolk sacs have been added to Fig. 4D. The het yolk sac vasculature is similar to that of the wild-type. Figs. 5 and 6 provide more detailed information on the evolution of vascular phenotypes following the switch to exclusive expression of EPHB4 F867L in the vasculature at E13.5. In summary, the first event is the back-filling of lymphatics with venous blood that likely arises due to the breakdown of lympho-venous valves. This is first observed in some embryos at E16.5 (images at E17.5 are shown in Fig. 5). At E17.5, apoptotic blood vascular endothelial cells are readily identified in the skin and brain, and there are clear signs of hemorrhage in these organs. Also, in the skin, collagen IV accumulates within blood vascular endothelial cells. By E18.5, blood vascular hemorrhage is more severe in the skin, and lymphatics are absent. We have reported these phenotypes previously in EPHB4 loss and RASA1 loss mutant embryos.

Wild-type embryos should be shown in Figure 4C.

Response: Wild-type embryos are now shown in the new Fig. 4E.

For the mice with global mutation plus endothelial-specific mutation, are there any phenotypes that resemble VOGM? For example, are there any arteriovenous malformations or vessel dilations?

Response: There are no phenotypes resembling VOGM, as all endothelial cells in this model are switched to express EPHB4 F867L alone, and all will eventually undergo apoptosis. In contrast, human VOGM lesions will arise from the combination of the inherited mutated EPHB4 allele plus a somatic "second hit" mutation in the inherited wild-type allele occurring in a single or a small fraction of endothelial cell(s). Thus, modeling EPHB4-related VOGM in mice would require disrupting the floxed allele in EphB fl/F867L Cdh5ert2cre mice in a single or very small number of endothelial cells. Neighboring endothelial cells without second-hit mutations would rescue those with second-hit mutations, perhaps through provision in *trans* of collagen IV (rescue from anoikis). For more than a decade, the King lab has attempted without success to generate mouse models of

EPHB4 and RASA1 gene disruption only in low numbers of cells. Simply put, the existing models of EPHB4 and RASA1-deficiency are the best currently available.

What does the cerebrovasculature look like? This model is interesting, but the data presented appears to be preliminary. Phenotypes should be assessed after activating Cre at different stages to better understand when EPHB4 might be contributing to VOGM-like phenotypes.

Response: Please see the comment about the cerebrovasculature above. Yes, we have observed apoptotic death of cerebrovascular endothelial cells and brain hemorrhage in E17.5 EphB fl/F867L Cdh5ert2cre embryos following tamoxifen administration at E13.5. This data is shown in new Figs. 5 and 6. Concerning additional times of tamoxifen administration, please note that blood vascular phenotypes arising from EPHB4 or RASA1 functional loss are restricted largely to the period of developmental angiogenesis. Disruption of either gene past E15.5 is without consequence in the blood vasculature, except under conditions of active angiogenesis (retinal angiogenesis in newborns, tumor angiogenesis in adults) or in the case of maintenance of venous valves (PMID: 35015735; 29024832; 26969842; 28530642)

The hemorrhage phenotype is not that informative.

Response: As indicated above, the current model that we have is the best available. Use of this model has clarified that endothelial cell-restricted expression of EPHB4 F867L in mice produces the same phenotype as EPHB4 loss.

Minor Comments:

1) What is the mutant Itgb1 Δ 10 mRNA that was injected in Itgb1a/b depleted zebrafish? Is it similar to the two damaging variants identified in two separate probands?

Response: The mutant Itgb1 Δ 10 mRNA was the 10 base pair deletion 9 (del GTCCTATTAA, p. S785fs) identified in KVOGM89-1(Supp Table 12 and Supp Figure 4). Please see comment above re: ITGB1.

2) The title of Table 1 indicates 114 VOGM cases, but in the table itself, there are 90 cases listed. This should be clarified.

Response: We have collected 114 VOGM cases in total, including 13 duos, 11 singletons and 90 trios (shown in Supp Table 1). *De novo* mutations were called in 90 trio cases only, as Table 1 indicated the enrichment of *de novo* mutations. The title has been changed to reflex 90 trios.

3) In Supplementary Figure 7C graphs, the x-axis mutant label is incorrect. This should be itgb1a/b mutants and not acvr1. Also, in the figure text for panel B and C, 'eph4a/b-depleted' should be replaced by the correct depleted zebrafish model.

Response: There were several mistaken labels in the original Figure. The data have been split across revised Supplementary Figure 8 and revised Supplementary Figure 9.

4) In the 3rd paragraph of the main text under "VOGM risk genes converge in growth factor-regulated Ras/ERK/MAPK signaling networks in developing cerebral endothelial cells" section, it should be referencing Fig. 3D instead of Fig. 3E.

Response: Fig. 3D has been referenced in the 3rd paragraph in this section.

5) In the 1st and 2nd paragraph of the main text under "Variants in ACVRL1 and other mutation-intolerant Mendelian vascular disease genes", it should be "R-hsa" instead of "R-has".

Response: "R-has" has been replaced by "R-hsa" in the 1st and 2nd paragraph of this section.

6) In Supplementary Figure 3A, it would be helpful to show the pedigree for clarity.

Response: The pedigree of this family was shown in original Figure 2 panel C, but the pedigree information has also been added in the revised figure legend (Supp. Figure 4A).

7) Can the authors find a different way to refer to the scRNA-seq modules? Referring to color isn't really helpful. Is there a diagram of the relationship of the modules that includes these colors?

Response: It is a convention in reporting WGCNA results to assign modules names based upon colors selected alphabetically from the 657 predefined colors supported in R (PMID:19114008). In order to both aid the readability and maintain the expected convention, we have added additional descriptors to the WGCNA modules such as 'the PCW 37 "Midnight Blue" module are enriched for those involved in Focal Adhesion-PI3K-Akt-mTOR Signaling Pathway (WP:3932; $P = 6.73 \times 10^{-10}$, 6.7-fold enrichment), including Positive Regulation of Vascular Development (GO: 1904018; enrichment 11.4-fold; $P = 7.95 \times 10^{-8}$) and Regulation of Cell Migration (GO: 0030334; enrichment 4.4-fold; $P = 5.94 \times 10^{-6}$; Fig. 3C)'.

8) Was the normal and arteriovenous malformation scRNA-seq data used from the Winkler et al. paper or just the normal dataset? This was not clear.

Response: Only the ~55,000 cells of the adult healthy vascular and perivascular regions were examined in the initial study. Please see below for our additional analysis regarding healthy vs. diseased tissue enrichment. We have completed a similar analysis for the diseased tissue which is included in **Extended Data Figure 2** below.

Was there a difference in enrichment for VOGM disease genes in the normal versus malformed vasculature?

Response: We have conducted the same modular analysis on 101,317 single cell transcriptomes of the malformed vasculature finding 23 distinct gene co-expression modules. Four modules were observed to be significantly enriched for our disease risk gene lists. Modules 12, 16, and 20 exhibited robust enrichment across multiple lists. Module 12 was significantly enriched for expression in FBMC cells while Module 16 was significantly associated with PC cells. Ontologically, Module 12 was defined by extracellular organization, PI3K-AKT-MTOR signaling, and immune/inflammatory response signaling. Module 16 was defined by WNT signaling and the development of the vasculature. Module 20 was primarily defined by neuronal death and was enriched only in CCM (**Extended Data Figure 1**).

Extended Data Figure 1: Modular analysis of 101,317 single-cell transcriptomes of the malformed human vasculature. A) Modular enrichments for each gene list. VOGM: Vein of Galen Malformation. AVM: Arteriovenous Malformation. AVM_VOGM: Combined AVM_VOGM list. pVOGM: Experimentally-determined possible Vein of Galen Malformation. CCM: Cerebral Cavernous Malformation. MMD: Moyamoya Disease. CHD: Congenital Heart Disease. Significant values are printed within their respective cell. **B)** Relative expression of gene modules per cell type. Asterisk indicates significant positive association of module gene expression in specified cell type for modules 12, 16, and 20. **C)** Gene ontology reports for modules 12, 16 and 20. Biological processes and Wikipathway enrichment is displayed. Vertical green bar indicates threshold for statistical significance. Module 20 has no significantly-associated Wikipathway terms.

Reviewer #3 (Remarks to the Author):

Zhao et al identify new risk genes for vein of Galen malformations (VOGMs), which are severe congenital brain arteriovenous malformations. Using trio-based WES data of a large VOGM cohort of 310 proband-family exomes, they identify pathogenic variants in *RASA1*, *EPHB4*, *ACVRL1*, *PTNPN11*, and *NOTCH1*. Integrated bioinformatic analysis of human cerebrovasculature single-cell transcriptomes to define co-expression modules, cell types, and biological pathways show enrichment of VOGM genes in endothelial cells, and reveals enrichment of pathways related to growth factor-regulated Ras/MAPK/ERK signaling in VOGM. Finally, some of the candidate genes were validated in vivo by characterizing vascular phenotypes in zebrafish or mouse carrying a VOGM gene mutation.

The first part of the study is based on an impressive patient cohort and provides highly valuable insight into genetic heterogeneity of and risk genes for VOGM. However, the in vivo validation of VOGM genes is too superficial to provide mechanistic understanding of disease pathogenesis, or to extend our current understanding of the role of these genes in regulating normal vascular development. To my mind the data would be stronger if focused on the genetic findings. This would also allow improving the clarity of the presentation of the clinical data. Alternatively, the authors should provide more in-depth analysis of disease pathogenesis by focusing on selected genetic mutation(s), taking into account that all of the identified VOGM risk genes have already been extensively studied in endothelial cells and in the context of vascular development and disease.

Response: We thank the reviewer for the positive feedback and valuable critique. We have addressed each comment below in a point-by-point manner.

Major comments:

1. Can the authors provide direct evidence for abnormal activity of Ras/MAPK/ERK in endothelial cells from VOGMs, to justify the statement in the title? For example, by staining of tissue biopsies, or WB analysis of patient-derived ECs?

Response: Unfortunately, we did not have banked tissue biopsies from patients to perform this experiment.

2. Methods such as Module and GO analysis, Reactome pathway analysis, and STRING analysis can provide valuable insight and a basis for generating hypotheses, but the results should be interpreted with caution and confirmed using other methods. Here the authors present several figures with such data, without additional validation of the conclusions drawn from these data.

Response: To further investigate and validate the findings obtained from Module and GO analysis, Reactome pathway analysis, and STRING analysis, we expanded our research by examining cell-type enrichment using two single-cell RNA sequencing (sc-RNAseq) transcriptomic atlases. This analysis revealed an enrichment of VOGM risk genes specifically in endothelial cells. Additionally, gene co-expression modules constructed from these sc-RNAseq datasets demonstrated an enrichment of VOGM risk genes as well as Ras signaling genes during fetal development. To test the hypothesis derived from these results, we conducted experiments using *EphB4* mutant mice. Mice expressing a VOGM-specific *EPHB4* kinase-domain missense variant exhibited disrupted developmental angiogenesis and impaired hierarchical development of angiogenesis-regulated arterial-capillary-venous networks. While we acknowledge the importance of further functional investigations, we believe that conducting extensive functional work is beyond the scope of our paper, which focuses primarily on the human genomic aspects of VOGMs.

3. The value of using two powerful single-cell RNA seq datasets in the study is unclear. The main conclusion is that ECs express genes associated with VOGM, but this is already known from previous studies. All of the identified VOGM risk genes have been extensively studied in endothelial cells and in the context of vascular development and disease. Additional data obtained from the module analysis is interesting but suggestive only.

Response: The purpose of the three single-cell analyses is to predict broader networks of pathomechanistic features related to the genes identified in the patient genomic analysis. These modules are created by

identifying genes with similar expression patterns across various cell types and developmental timepoints, and therefore predicted to be involved in the same biological processes. Disease risk gene lists can then be examined across each module using a Bonferroni-corrected hypergeometric enrichment analysis to identify modules enriched for the presence of specific gene lists. These enriched modules can then be analyzed for their respective ontological profiles, developmental time point correlations, and cell type enrichments. Regarding the color names in the weighted gene co-expression network analysis (WGCNA), it is a long-standing convention in reporting WGCNA results to assign module names based upon colors selected alphabetically from the 657 predefined colors supported in R. To aid readability while maintaining convention, we have added additional descriptors to the WGCNA modules such as “the PCW 37 Midnight Blue module”.

4. Figure 2, Zebrafish phenotypes: do all images represent the same depth/part of the brain, for example in Figure 2E? I note missing flow in the smaller vessels in the central arteries. Is it possible that vein enlargement is secondary to impaired/disturbed flow?

Response: Regarding Figure 2, it is important to note that all the displayed images in the figure represent maximum intensity projections of confocal stacks taken between the level of the PCS (ventral most slice) and the very dorsal most aspect of the cranial vasculature (i.e., the dorsal midline junction). This represents effectively all the cranial vasculature of the zebrafish at this stage.

The reviewer astutely observes the absence of flow in the smaller vessels of the central arteries. During these stages of developmental stages the central arteries typically do not carry flow. This is not considered abnormal, as these vessels in early stages of formation will fill with blood only at later stages of development.

The reviewer raises an additional intriguing point regarding the possibility that vein enlargement observed in the cases of EphB4a/b and Acvr1 could be a result of impaired or disturbed flow. While we acknowledge this as a potential mechanism, we do not speculate on whether it is the specific underlying cause in our study.

Better characterization of the phenotypes, including a full developmental time course, would be needed to understand the mechanism of pathogenesis.

Response: We agree that a developmental time course could help provide a better understanding the mechanism of pathogenesis. However, such a developmental time course study will require creation of multiple new mouse lines, each requiring characterization. We therefore suggest that these timely and costly experiments lie outside of the scope of the current study focused largely on human genomics. However, developmental studies will be important topics of future grant proposals and experiments in our lab.

5. Page 11: “..the possible requirement of EPHB4 kinase activity for normal vascular development is unknown”. Kinase inactivating mutations in EPHB4 were previously described in non-immune hydrops fetalis and lymphedema (PMID: 27400125). The authors should discuss these findings in the context of theirs.

Response: We have now clarified that we are referring to blood vascular abnormalities. PMID: 27400125 is now quoted in the Results section in the context of lymphovenous valve development.

6. Figure 4, VOGM-specific EPHB4 mutant mouse: The authors focus on cutaneous lesions, which they mention were not observed in the VOGM patients. Do these EPHB4 mutants also recapitulate lesions in the brain?

Response: We have provided a more extensive characterization of the inducible EPHB4 F867L mouse model in the revised submission (Figs. 5 and 6). The model phenocopies inducible EPHB4 and RASA1 loss mutant models.

In brief, following the induced switch to exclusive expression of EPHB4 F867L in the vasculature at E13.5, the first event is the back-filling of lymphatics with venous blood that likely arises as a result of the breakdown of lymphovenous valves, as reported previously in a vascular-specific inducible EPHB4 loss model (PMID: 27400125). This is first observed in some embryos at E16.5 (images at E17.5 are shown in Fig. 5). At E17.5, apoptotic blood vascular endothelial cells are readily identified in the skin and brain, and there are clear

signs of hemorrhage in these organs. Also, in the skin, collagen IV accumulates within blood vascular endothelial cells. By E18.5, blood vascular hemorrhage is more severe in the skin, and lymphatics are absent. We have reported these blood vascular phenotypes in vascular-specific inducible EPHB4 loss and RASA1 loss mutant embryos previously.

Figure 4E: Is this hemorrhage, or blood-filled lymphatic vessels?

Response: The image presented in the original version of the paper was hemorrhage, but the reviewer is correct that blood-filled lymphatics are a feature of this model at E16.5 to E17.5— see response above.

Increased pERK should be shown in ECs (as I understand now shown in total liver lysate). Please show pAKT as well (also enriched in VOGM based in module analysis in Figure 3).

Response: We have had difficulty demonstrating increased MAPK (or AKT) activation in the vasculature of EPHB4 F867L embryos by immunostaining of tissue sections. Our current understanding of EPHB4 and RASA1 signaling would favor dysregulated Ras-MAPK activation as the most likely cause of the vascular defects. However, we have softened our proposals concerning downstream signaling and focused on the validation of our novel genomic findings. As noted above, we have provided further characterization of the model to show that it phenocopies EPHB4-null and RASA1-null models (new Figs. 4-6).

Other comments:

1. Graphical abstract has too much detailed information and does not summarize key findings of the study.

Response: We have simplified the figure and summarized key findings of the study.

2. Figure panels should be labelled in the order they are referred to in the text (for example Figure 2E vs 2F)

Response: This figure has been reorganized. Figure panels are now labelled in the order in which they are referred to in the text.

3. Page 14: "...define developing endothelial cells as a critical spatio-temporal locus of VOGM pathophysiology. This appears to be different from other cerebral vascular anomalies such as Moyamoya disease and cerebral cavernous malformations". To my knowledge, CCM gene mutations in ECs drive cerebral cavernous malformations. Please comment.

Response: We appreciate the clarification regarding the role of CCM genes in endothelial cells and their association with cerebral cavernous malformations (CCMs). We acknowledge the studies (PMID: 33234067; 35771000) demonstrating widespread expression of CCM genes in ECs of CCM lesions and the development of CCM-like neurovasculature lesions in mice with EC-specific deletion of CCM genes. Prevention of lesion formation through endothelial-specific loss of Mekk3, Klf2, or Klf4 (PMID: 27027284) further supports the critical involvement of endothelial cells in CCM pathogenesis.

In our analysis, we included the CCM genes PIK3CA, MAP3K3, KRIT1, CCM2, and PDCD10. Although we did not find significant enrichment of CCM gene mutations in ECs in Figure 3 (p value = 0.0692), we further explored the expression of CCM genes in a developing cerebrovasculature single-cell RNA dataset. Interestingly, we found significant enrichment of CCM genes in Mitotic Endothelial Cells (P = 0.0002511886) in this dataset, even after multiple testing correction. This suggests that the original dataset used, which focused on adult cerebrovasculature, might have different expression profiles compared to the developing cerebrovascular dataset. Moreover, inclusion of only five genes in the CCM gene list and potential batch effects between different datasets may significantly impact the enrichment results for this gene list.

Considering the limitations of the small gene set used in the analysis, we agree that the statement "This appears to be different from other cerebral vascular anomalies such as..." was misleading and have removed it from our text. Thank you for bringing this to our attention.

4. Supplementary Figure 1: Duplicated image KVOGM 65-1 and KVOGM 70-1.

Response: We have corrected this in Supplementary Figure 1.

5. Supplementary Figure 7C: Figure legends states ITGB1, figure acvr1

Response: This original figure included several instances of mislabeling. The corrected versions can now be found as Supplementary Figure 8 and Supplementary Figure 9. Please also see comment above re: ITGB1.

6. Cdh5-ert2cre – should it be Cdh5-creERT2?

Response: We have always used ert2cre in our publications.

7. Especially in the results, it was not always clear which data are novel, and what has been shown previously. One example is below, but authors are recommended to check the entire manuscript carefully and refer to previous findings where applicable: Page 11: "Based on our human genetic (Fig. 1) and integrative genomic results (Fig. 3), we hypothesized that genetic inactivation of a kinase-dependent, EPHB4-regulated RASA1 signaling mechanism in endothelial cells causes constitutive Ras-MAPK activation and disruption of VEGF-associated developmental angiogenesis in vivo." As the authors discuss later, EPHB4/RASA1 deficiency has been shown to result in increased Ras/MAPK activation and the relevant studies should be referred to in this context.

Response: Please see above. The key phrase here is "kinase-dependent," which has not been previously demonstrated in vasculature.

REVIEWER COMMENTS

Reviewer #1 (Remarks to the Author):

The first manuscript by Zhao et al., presented exciting data highlighting novel mechanistic insight into VOGM, however, some major issues remained unresolved.

Unfortunately, the revised manuscript falls short from addressing these issues and even presents a major setback in comparison to previous manuscript regarding mechanistic EphB4 activity in developmental angiogenesis.

Major issues :

-MAPK/ERK signaling analysis has been dropped.

-The itg1a/b analysis has been dropped.

-Zebrafish imaging remains highly pixelized and of poor quality, one would expect a better analysis at this level of publication.

-The reviewer is slightly puzzled by novel data presented in Figure 1. Authors re-adjust their WB by showing weaker exposures of EPHB4 blots, and Act B is now stable throughout. Are these separate Act B loadings? Is this quantified by comparing two separate loadings?

Reviewer #2 (Remarks to the Author):

The authors have completely responded to the previous reviews and the manuscript is greatly improved. The findings are novel and interesting.

I have only one minor comment: The number of embryos assessed for the representative images of EPHB4 mouse experiments should be indicated.

Reviewer #3 (Remarks to the Author):

In the revised manuscript, the authors have aimed to strengthen the mechanistic understanding of VOGM disease pathogenesis. Specifically, the authors have conducted the characterization of a mouse model with inducible VOGM-specific kinase-inactivating EPHB4 mutation. They show that these mice recapitulate EPHB4 loss of function phenotype, exhibiting disrupted developmental angiogenesis and lympho-venous valve formation. The novelty of the findings compared to previous studies lies in the demonstration of kinase-dependent function of EPHB4 in developmental angiogenesis. However, the model provides limited insight into the mechanisms of VOGM, likely, as discussed by the authors in the rebuttal letter, because the model does not recapitulate human sporadic disease.

Overall, the primary strength of the manuscript lies in providing valuable insights into the genetic heterogeneity of and identifying risk genes associated with VOGM. However, the title does not accurately represent the findings, considering that the authors have not been able to directly show dysregulated Ras/MAPK/ERK activity in the endothelium of human VOGM or the mouse model, but this conclusion is based on analysis of modules of VOGM-enriched genes (where other pathways such as PI3K-AKT are also identified) and previous literature showing the role of the identified risk genes as regulators of Ras.

Minor comments:

I suggest again that the authors consider using nomenclature for the Cdh5-CreERT2 line according to the original publication and MGI nomenclature, at least in the methods section. PMID: 20445537, <https://www.informatics.jax.org/allele/MGI:3848982>

Reviewer #1 (Remarks to the Author):

The first manuscript by Zhao et al., presented exciting data highlighting novel mechanistic insight into VOGM, however, some major issues remained unresolved.

Unfortunately, the revised manuscript falls short from addressing these issues and even presents a major setback in comparison to previous manuscript regarding mechanistic EphB4 activity in developmental angiogenesis.

Major issues:

-MAPK/ERK signaling analysis has been dropped.

Response: We appreciate the reviewer's attention to the MAPK/ERK signaling analysis in our paper, as this pathway holds significant relevance to our research. As explained in our first response, technical limitations preclude us from making definitive statements on changes in the magnitude or duration of MAPK signaling in EPHB4 mutants. Initial Western blot analysis of increased MAPK activation in mutants proved inconsistent when larger numbers of embryos were examined. In addition, immunostaining analysis of mutant tissue sections gave inconsistent results. We suspect that the change in endothelial cell MAPK activation in mutants is highly restricted, both temporally and spatially. Detection of such changes will require an improvement in existing technologies, specifically an ability to identify phospho-MAPK in optically-cleared mid to late gestation whole mount embryos at single-cell resolution.

Despite the above limitations, we have taken the reviewer's feedback into consideration and have provided a significant amount of new data to elucidate the mechanism and functional consequences of the Ephb4 F867L mutation through an extended analysis of EphB4 fl/F867L Cdh5-CreERT2 embryos, another novel mouse line of our creation. These results show unequivocally that the EphB4 F867L mutants phenocopy EphB4 and Rasa1 null mutants, highlighting a requirement of EphB4 kinase function for developmental angiogenesis and lymphovenous valve formation (new **Figures 6 and 7**). Moreover, the findings validate for the first time in a mammalian system the pathogenicity of a VOGM-associated EPHB4 missense mutation and shed insight into the likely requirement of a two-hit mechanism in VOGM pathogenesis. Based on these results, we do not feel these results are a "major setback" to our initial submission. Furthermore, they do not detract from the core objective and message of our paper, which remains to provide the most in-depth analysis to date of the genetic architecture of VOGM in the largest cohort ever assembled. However, we agree with the recommendation that we de-emphasize and further tone down the potential role of Ras/MAPK signaling upregulation in disease pathogenesis. We have made these important changes in the title and throughout the manuscript and figures, including the model schematic.

As an interesting aside, the pathogenic *de novo* p.Tyr63Cys variant in *PTPN11*, encoding the non-receptor tyrosine phosphatase SHP2 (**Table S12** and **Fig. S4**) identified in proband KVOGM23-1, has been reported in an unrelated patient Noonan syndrome type 1 (OMIM# 163950), which features a broad spectrum of congenital heart defects and other systemic vascular lesions^{77,78}. Similar to other N-SH2 domain 'blocking loop' variants, p.Tyr63Cys has been shown experimentally to disrupt SHP2 autoinhibition and cause constitutive activation of Ras signaling^{79,80}. The paralogous p.Asp61Gly variant in SHP1 also has been shown to potentiate Ras/ERK/MAPK signaling by interfering with phosphorylation of GAB1, an adaptor protein recruiting RASA1 to receptor tyrosine kinases^{79,81,82}. Coupled with the known role of RASA1-EPHB4 in Ras/MAPK signaling, these findings suggest that experiments examining

effects of VOGM mutations on Ras and other downstream signaling pathways may be fruitful lines of investigation for future stand-alone biochemical papers.

-The *itg1a/b* analysis has been dropped.

Response: We appreciate the reviewer's feedback concerning the *itgb1a/b* analysis in our paper. In our initial submission, we decided to functionally validate ITGB1 because of the compelling genetics and previous research that established the essential role of *Itgb1* in embryonic vasculature patterning and postnatal vascular remodeling (PMID: 25752958; 17984225; 20152176). In our original functional experiments, we observed increased AV contacts in *itgb1a/b* mutants, which prompted us to include the data. However, one reviewer was unconvinced about the AV contact phenotype observed in *itgb* mutant fish and suggested the use of blinding to score phenotypes. We also chose to conduct a scrambled gRNA injection control. These steps were taken to increase the rigor and reproducibility of our study. Despite maintaining the observed trends, the analysis revealed an effect size for *itgb1a/b* just below the significance threshold, as shown in Figure below:

We nonetheless recognize the potential importance of these findings suggesting ITGB1 mutations as a rare cause of VOGM, and therefore have left in the genetic data that reports the identification of the variants. Further functional validation, especially after the detection of additional variants as exome sequencing continues with new patients, will be continued topics of investigation for our lab.

-Zebrafish imaging remains highly pixelized and of poor quality, one would expect a better analysis at this level of publication.

Response: We appreciate the reviewer's noting this formatting issue. Based on the reviewer's suggestions, we have made necessary adjustments to improve the pixelization and overall quality of the zebrafish images and restructured the figure layout to present the imaging results in a more organized and coherent manner to enhance the clarity and presentation of our findings (new **Figure 3**).

-The reviewer is slightly puzzled by novel data presented in Figure 1. Authors re-adjust their WB by showing weaker exposures of EPHB4 blots, and Act B is now stable throughout. Are these

separate Act B loadings? Is this quantified by comparing two separate loadings?

Response: In the previous review, the reviewer commented several times that our blots were over-exposed. We therefore provided weaker exposures of the relevant blots and performed quantitative analyses of these weaker exposures, as the reviewer had requested. Concerning the comment about ActB now appearing stable throughout the time course of the protein decay experiments (Fig. 1F), this reflects not weaker exposure or use of ActB exposures from a different experiment, but rather that our original submission mistakenly provided blots of SLC25A9 decay for the WT and F867L samples. For reference, below is the response we provided to this point in the first rebuttal.

“Fig. 1F. Weaker exposures of blots are provided and quantitated. In the previous version of the manuscript, we inadvertently provided images of SLC25A9 decay instead of actin B decay for the WT and F867L transfection experiments. The correct images are now provided. In summary, there is no obvious difference in the rate of decay of EPHB4 kinase mutants compared to WT EPHB4.”

Fig. 1F - SLC25A9 and ActB probes

In case there is any confusion on this point, we have provided with this resubmission images of entire ECL blots showing rates of decay of SLC25A9 and ActB for each EPHB4 variant above.

Reviewer #2 (Remarks to the Author):

The authors have completely responded to the previous reviews and the manuscript is greatly improved. The findings are novel and interesting.

Response: We appreciate the positive feedback and thank the reviewer for his/her time and comments.

I have only one minor comment: The number of embryos assessed for the representative images of EPHB4 mouse experiments should be indicated.

Response: This information is now provided in Figure legends at the appropriate points.

Reviewer #3 (Remarks to the Author):

In the revised manuscript, the authors have aimed to strengthen the mechanistic understanding of VOGM disease pathogenesis. Specifically, the authors have conducted the characterization of a mouse model with inducible VOGM-specific kinase-inactivating EPHB4 mutation. They show that these mice recapitulate EPHB4 loss of function phenotype, exhibiting disrupted developmental angiogenesis and lympho-venous valve formation. The novelty of the findings compared to previous studies lies in the demonstration of kinase-dependent function of EPHB4 in developmental angiogenesis. However, the model provides limited insight into the mechanisms of VOGM, likely, as discussed by the authors in the rebuttal letter, because the model does not recapitulate human sporadic disease.

Response: We agree, but as outlined in the original rebuttal letter, these are the best models currently available to the field. Development of a VOGM mouse model that fully recapitulates the human disease would constitute an extensive and entirely separate study. We have been attempting to develop such a model using multiple genetic approaches for more than a decade without success. We hope that the reviewer can agree that inclusion of such a model (that does not yet exist) is beyond the scope of the current manuscript, whose main focus is to provide the most in-depth analysis to date of the genetic architecture of VOGM in the largest cohort ever assembled.

Overall, the primary strength of the manuscript lies in providing valuable insights into the genetic heterogeneity of and identifying risk genes associated with VOGM. However, the title does not accurately represent the findings, considering that the authors have not been able to directly show dysregulated Ras/MAPK/ERK activity in the endothelium of human VOGM or the mouse model, but this conclusion is based on analysis of modules of VOGM-enriched genes (where other pathways such as PI3K-AKT are also identified) and previous literature showing the role of the identified risk genes as regulators of Ras.

Response: We appreciate the reviewer's positive evaluation of our manuscript's strength in providing valuable insights into VOGM genetic heterogeneity and in identifying risk genes associated with VOGM. We also acknowledge the concern regarding accuracy of the title in fully representing our findings. We therefore propose a revised title that better reflects the core contributions of our research: "Mutation of key signaling regulators of cerebrovascular development in vein of Galen malformations".

Our revised title emphasizes the comprehensive nature of our study, which explores the genetic heterogeneity of VOGM using an integrative genomic approach and delves into the mechanisms causing disrupted developmental angiogenesis in this condition. We believe this revised title accurately highlights our findings advancing understanding of VOGM genomics, validating new VOGM genes, and providing initial insights into mechanisms of VOGM pathogenesis. Further cellular and molecular details of VOGM pathogenesis will be subjects of additional papers.

Minor comments:

I suggest again that the authors consider using nomenclature for the Cdh5-CreERT2 line according to the original publication and MGI nomenclature, at least in the methods section. PMID: 20445537, <https://www.informatics.jax.org/allele/MGI:3848982>

Response: We now use the term Cdh5-CreERT2 throughout.